# Multi-day rhythms modulate seizure risk in epilepsy

Maxime O. Baud[1,2,3,4], Jonathan K. Kleen[1], Emily A. Mirro[5], Jason C. Andrechak [6], David King-Stephens[7], Edward F. Chang[8] & Vikram R. Rao[1]

Epilepsy is defined by the seemingly random occurrence of spontaneous seizures. The ability to anticipate seizures would enable preventative treatment strategies. A central but unresolved question concerns the relationship of seizure timing to fluctuating rates of interictal epileptiform discharges (here termed interictal epileptiform activity, IEA), a marker of brain irritability observed between seizures by electroencephalography (EEG). Here, in 37 subjects with an implanted brain stimulation device that detects IEA and seizures over years, we find that IEA oscillates with circadian and subject-specific multidien (multi-day) periods. Multidien periodicities, most commonly 20–30 days in duration, are robust and relatively stable for up to 10 years in men and women. We show that seizures occur preferentially during the rising phase of multidien IEA rhythms. Combining phase information from circadian and multidien IEA rhythms provides a novel biomarker for determining relative seizure risk with a large effect size in most subjects.

[1] Department of Neurology and Weill Institute for Neurosciences, University of California, San Francisco, CA 94143, USA. [2] Department of Neurology, University Hospital Geneva, Rue Gabrielle-Perret-Gentil 4, 1205 Geneva, Switzerland. [3] Wyss Center for Bio and Neuroengineering, 1202 Geneva, Switzerland. [4] Sleep-Wake-Epilepsy-Center, Department of Neurology, Inselspital, University of Bern, 3010 Bern, Switzerland. [5] NeuroPace, Inc., 455N. Bernardo Ave, Mountain View, CA 94043, USA. [6] Department of Chemical and Biomolecular Engineering, University of Delaware, Newark, DE 19716, USA. [7] Department of Neurology, California Pacific Medical Center, San Francisco, CA 94115, USA. [8] Department of Neurological Surgery and Weill Institute for Neurosciences, University of California, San Francisco, CA 94143, USA. Correspondence and requests for materials should be addressed to M.O.B. (email: maxime.baud.neuro@gmail.com)

Daily to monthly patterns in seizure occurrence have been described since antiquity[1] but only recently quantified, revealing circadian[2–4], and cluster organization[5,6]. The existence of such patterns suggests that brain activity is regulated over long timescales. Despite recent progress in using features of interictal brain activity to forecast imminent seizures[7–10], controversy remains regarding the relationship between seizures and interictal epileptiform discharges (e.g., spike-waves, polyspikes, and fast oscillations)[11–13]. The rate of these pathological discharges (here termed interictal epileptiform activity, IEA) fluctuates over time and may increase or decrease before seizures. This suggests that IEA and seizures are dissociable but influenced by a common process[14], one that may be periodic. Observation of such a process requires chronic recordings of brain activity capturing multiple cycles[15].

Recently, an FDA-approved closed-loop implantable brain stimulator for detecting and treating seizures (NeuroPace, Inc. RNS® System, hereafter referred to as 'RNS System') has afforded an unprecedented opportunity to monitor human brain activity with intracranial recordings continuously over many years. The RNS System involves a programmable neurostimulator connected to intracranial electrodes recording neural activity at the seizure focus or foci. Storage of raw EEG on the device is limited, but customizable algorithms are used to record hourly counts of epileptiform discharges (IEA) and timestamps of seizures. These data sets are well-suited for analysis of IEA rhythms at long timescales. Here, using wavelet transform to decompose IEA time-series, we identify multidien rhythms[16] with period lengths that are variable across, but relatively stable within, male and female subjects over years of recording. Seizures occur preferentially on the days-long up-slope of the multidien rhythm, independent of period length. Specific circadian timing of seizures is more variable across subjects, but we show that, at the individual level, multidien and circadian IEA rhythms are co-determinants of seizure risk.

## Results

**Subjects**. We studied 37 subjects (22 males; age range 22–58) with epilepsy who had been implanted with the RNS System (Fig. 1a) for approved clinical indications. Lead placements included mesial temporal ($N = 23$) and neocortical regions ($N = 14$) and were bilateral in 20 subjects (Supplementary Table 1). Recording durations were 3 months to 9.9 years (median: 2.3 y). For this study, IEA is defined as hourly rates of detections of epileptiform discharges using subject-specific algorithms designed by clinicians (Supplementary Fig. 1a).

**Characterization of circadian and multidien rhythms of IEA**. We formatted IEA rate (Fig. 1b, c) into continuous time-series and applied wavelet transform to resolve component rhythmicity of the IEA (signal processing steps depicted in detail in Supplementary Figs. 2 and 3). Individual subjects showed clear circadian variation (Fig. 1d, h), and multidien rhythms were also apparent in daily averages of IEA plotted over long periods (Fig. 1e, i, j, Supplementary Fig. 4). Spectral decomposition (Fig. 1f, g) revealed the expected peaks in ultradian (12 h) and circadian (24 h) rhythms[3,17,18], as well as longer periodicities (5.5–33 days) in most subjects (Fig. 2a, Supplementary Fig. 4). The median ratio of multidien to circadian peak amplitude was 1.4 (range 0.4–5.7; >1.0 in 27 subjects), suggesting that the multidien rhythm was as robust as the circadian rhythm in most subjects. The most common periodicities were 26–30 days ($N = 18$) followed by 20–22 days ($N = 16$, Fig. 2b). Intra-subject autocorrelation coefficients in the frequency-domain over time were above 0.5 for all ($0.72 \pm 0.07$, Supplementary Fig. 5a, b, e), reflecting relative

stability of these rhythms. Intra-subject correlation of periodograms derived from bihemispheric recordings was $0.93 \pm 0.05$ ($N = 8$, Supplementary Fig. 5c, d), suggesting that IEA rhythms in anatomically distinct seizure foci are co-regulated. All subjects exhibited a 12-h harmonic of the circadian rhythm and some subjects also had harmonics of multidien rhythms, for example with peaks at 7.5 and 15 days (Supplementary Fig. 6). Unsupervised clustering of the periodograms based on their coefficients for principal components (Supplementary Fig. 7) showed three patterns of multidien rhythms: (i) about weekly and biweekly ($N = 9$), (ii) about tri-weekly ($N = 12$), and (iii) about monthly ($N = 16$) peaks (Fig. 2a). This analysis was done mainly for visualization purposes, as there was no strong categorical separation of the data. Rather, the range of periods was a continuum, sometimes with two or three peaks in the same subject (Fig. 1, Supplementary Fig. 4). Subjects demonstrating these patterns did not differ significantly by region of seizure onset ($p = 0.87$, $\chi^2$-test). Male and female subjects showed a similar distribution of periodicities ($p = 0.87$, $\chi^2$-test, Fig. 2b). Multidien rhythms remained apparent during times when the stimulation function of the RNS System was disabled (Supplementary Fig. 4).

**Phase analysis of IEA peak in relation to circadian time**. To complement this power analysis with time information, a phase analysis of the circadian IEA rhythm revealed that the peak aligned consistently with a given hour in all subjects ($p < 0.001$, Omnibus test, Fig. 3). Unsupervised clustering showed three groups with peaks around 4:00 PM, 2:00 AM, and 6:00 AM, which may represent different IEA chronotypes[19] (Fig. 3). These findings are consistent with a prior study showing peak nocturnal occurrence of IEA independent of the region of seizure onset ($p = 0.14$, $\chi^2$-test)[3].

**Phase analysis of seizure timing relative to IEA rhythms**. Next, we investigated the relationship of seizures to the phase of the underlying circadian and multidien IEA rhythms in a subset of subjects ($N = 14$) for whom seizure detection by the RNS System was highly reliable (estimated ~2% false positives, see Methods section). Average seizure rate across these subjects ranged from one seizure every 17 days to 32 seizures per day, with a median total number of seizures of 325 (range 74–115,154). Circular statistics of seizure timing confirmed significant entrainment to circadian and multidien rhythms in 12 and 13 out of 14 subjects, respectively (Omnibus test, p-values in Fig. 4). Across subjects, the average phase-locking value (PLV, equal to the resultant vector length) was similar for circadian and multidien rhythms (Fig. 4a, b; $p = 0.63$, Wilcoxon test), suggesting that seizures were tied as strongly to a given phase of multidien rhythms as to circadian rhythms. However, the population of angles was significantly different ($p = 0.002$, Kuiper test), spanning from trough to peak of circadian rhythms and from up-slope to peak of multidien rhythms. Seizures were therefore coupled to multidien rhythms over a more narrow range of phases than to circadian rhythms. Circadian and multidien PLVs correlated weakly (Pearson $r = 0.36$, $p = 0.20$). To further explore the relationship between circadian and multidien timing, individual seizures were mapped on the circadian vs. multidien phase-space (see Methods section and Fig. 5), revealing uncorrelated grouping of seizures at preferred circadian and multidien phases independent of the underlying period (Fig. 5). Multidien PLVs inversely correlated with seizure rate (Pearson $r = -0.7$, $p = 0.005$, Supplementary Fig. 8). Thus, seizures are more tightly coupled to the preferred multidien phase in subjects with low or moderate seizure rates (the majority in this study) than in subjects with high seizure rates.

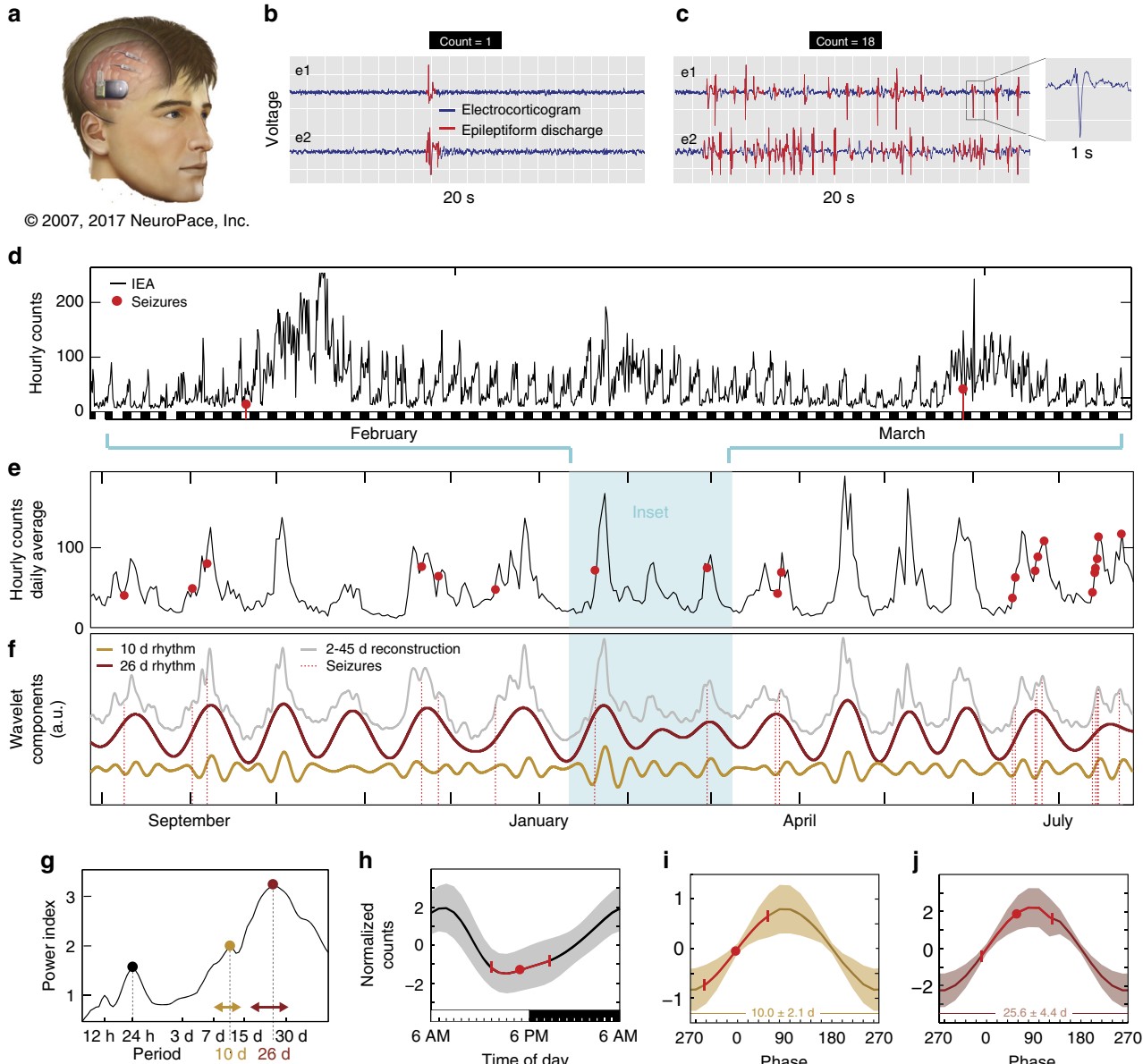

**Fig. 1** Representative subject demonstrating circadian and multidien rhythms in IEA, as well as preferential timing of seizures. **a** RNS System comprising cranially implanted neurostimulator connected to intracranial leads (image used with permission from NeuroPace, Inc.). **b** EEG showing a single-epileptiform discharge (spike) in channels corresponding to left (e1) and right (e2) hippocampal leads. **c** EEG recorded 1 week later at the same time of day showing higher count of epileptiform discharges, i.e., higher IEA. Inset magnifies one typical element to show waveform morphology. Hourly (**d**, cyan inset) and daily (**e**) fluctuation in IEA in one subject over 2 and 12 months, respectively. Red dots indicate times of seizure occurrence. **f** Wavelet decomposition revealing two component multidien rhythms with periodicities of 10 and 26 days. Combining all multidien wavelet coefficients reconstructs the daily IEA time-series (gray curve, 2–45 d, Pearson correlation $r = 0.93$, $p = 0$). **g** Corresponding periodogram showing ultradian (12 h), circadian (24 h), and multidien (10 and 26 d) peaks in periodicity. Period length displayed on the x-axis, and power index (square root of spectrogram power) on the y-axis. Horizontal double-arrows show span of corresponding wavelet coefficients included for (**f**) (peak period ± 33%). **h** Average normalized amplitude of the circadian rhythm as a function of time of the day showing phase preference of seizures near the trough at 5 PM ($n = 74$ seizures, mean ± SD in red, $p = 10^{-4}$, Omnibus test, see Methods section). Black and white rectangles (**d**, **h**) represent night (6PM–6AM) and day (6AM–6PM), respectively. **i**, **j** Average normalized amplitude of the 10 d and 26 d IEA rhythms as a function of their underlying phase (x-axis, full 360 degrees phase; y-axes have different scales). Seizures demonstrate phase preference for the up-slope of both rhythms (10 and 26 days, $n = 66$ seizures, mean ± SD in red, $p = 0.0002$ and $p = 0.002$, respectively, Omnibus test)

**Seizure risk modulation by circadian and multidien rhythms.** We estimated differences in effect size across subjects by calculating the risk of having a seizure at a given circadian or multidien phase relative to the risk of not having a seizure at the same phase. When combining phase information from the underlying circadian and multidien rhythms, we found small (risk ratio (RR) 1.2, 95% CI: 1.1–1.3) to very large effect sizes (RR 24.5, 95% CI: 3.4–175.9) in subjects with high and low seizure rates, respectively, and a large effect-size summary across subjects (unweighted RR 6.8, 95% CI: 3.1–15.1, Fig. 5). Although the relative modulation of seizure occurrence by circadian and multidien rhythms varied across subjects, the highest risk ratio was

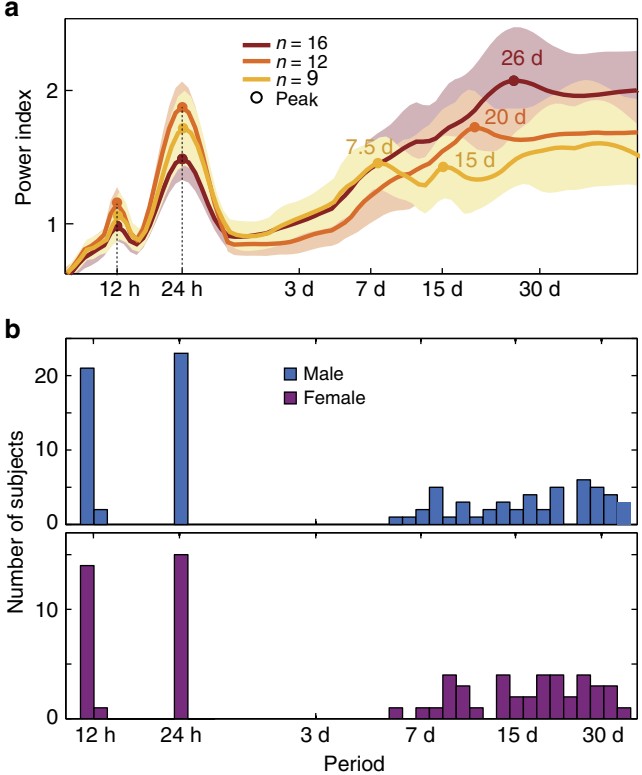

**Fig. 2** Periodograms and peaks of IEA rhythms. **a** Average periodograms across all subjects ($N = 37$) showing ultradian, circadian, and multidien peaks. For better visualization, unsupervised clustering across all subjects revealed three patterns: (i) about weekly-to-biweekly rhythm (peaks at 7.5 and 15 days, $N = 9$), (ii) about tri-weekly rhythm (peak at 20 days, $N = 12$), and (iii) about monthly rhythm (peak at 26 days, $N = 16$). Shading indicates $\pm 1$ SD. **b** Histograms showing the number of subjects with a peak in the periodogram at a given period. The distributions are similar ($p = 0.87$, $\chi^2$-test) in male ($N = 22$) and female ($N = 15$) subjects

found when the two critical phases were combined (Fig. 6). When multidien and circadian rhythms were both anti-phase, seizures were rare in 5 out of 14 subjects (S3, S5, S24, S30, and S33).

## Discussion

Our results reveal that, in addition to well-known circadian rhythms, IEA fluctuates with slower multidien rhythms that vary across subjects but are relatively stable within subjects over many years. Furthermore, seizures occur preferentially during narrow phases of these circadian and multidien rhythms. Thus, seizures are organized by underlying biological rhythms that operate over multiple timescales and jointly modulate seizure risk.

Previous applications of quantitative methods to chronic intracranial recordings have elegantly characterized distributions of seizure durations and inter-seizure intervals[5,20], established power-law relationships linking past and future seizures[6], and identified circadian and ultradian patterns[2–4]. One study using an autocorrelation method in the time-domain found cyclical patterns of IEA ranging in duration from weeks to 1 month in a limited number of subjects[4]. Here we took advantage of longer recordings and frequency-domain statistical analyses designed for the study of oscillations at any scale. Our study is distinguished by the elucidation of multidien rhythms in most subjects, often with greater magnitude than circadian modulation. This is remarkable as multidien rhythms were present covertly in all-comers, even though most did not have obvious periodicity of their epilepsy, underscoring the value of monitoring IEA as a biomarker of

disease activity. In comparison with previous contributions[4], the key insight from our work is that, across subjects with diverse focal epilepsies, seizure timing depends on the phase of the multidien rhythm, explaining how seizures tend to form clusters with long-range dependencies[5,6,20]. This phenomenon could only be elucidated with long timescale recordings of IEA and seizures, and the importance of using this wider temporal lens to view and anticipate seizure dynamics represents a major conceptual advance. Indeed, the time window of pre-seizure activity relevant for seizure prediction may be on the scale of days rather than hours as previously thought[10,21]. Overall, seizure occurrence was best explained by incorporating information about circadian and multidien rhythms. Reliable real-time seizure prediction will likely involve a combinatorial function of multiple features of an individual's epilepsy, including past and present seizure characteristics and short and long-term IEA trends. Multidien and circadian rhythms may be most predictive in subjects with a low or moderate seizure rate where phase preference is highest.

The data presented here, based on analysis of thousands of seizures, help reconcile conflicting evidence regarding the relationship between IEA and seizures. Previous studies have reported that IEA increases, decreases, or remains unchanged before seizures[4,11,22], and IEA trends after seizures are also variable[11,23,24]. Seizures preferentially occur during the rising phase of multidien IEA cycles, but, in a given subject, this could coincide with the peak or the trough of the circadian IEA cycle[4], perhaps explaining how shorter timescale studies, looking at hour-to-hour changes in IEA, could draw seemingly contradictory conclusions. Similarly, day-to-day changes in IEA may not explain seizure timing as well as the phase of the underlying slow oscillation. A major advantage of our study is that chronic recordings were made in ambulatory subjects under natural conditions, i.e., without tapering anticonvulsant medications, which is typical of acute inpatient recordings and known to affect IEA[24].

Our findings challenge the concept of a direct, generalizable relationship between IEA and seizures and favor a hypothesis that these epilepsy phenomena covary under differential influence of factors operating at multiple timescales. A slow permittivity variable was recently identified in an elegant mathematical model of epilepsy[25], and our results support the existence of an unidentified factor (or factors) regulating slow epileptic fluctuations[26], possibly through changes in brain metabolism[27] or circuit function[28]. Further analysis of the rise and decay kinetics of IEA fluctuations may be informative with regard to underlying biological mechanisms. We speculate that the seemingly independent circadian and multidien oscillators may in fact be co-modulated by hormonal, genetic, environmental[29], sleep-wake cycle[30], and behavioral factors[31]. Hormonal influence on seizures occurs in catamenial epilepsy[32,33], and one of the 15 female subjects had seizures related to menstrual cycles[32,33] with IEA cycling at 13 and 26 days. However, we observed similar rhythmicity in men, so catamenial cycling cannot explain our results.

This study has limitations. Our subjects, who have medically refractory focal epilepsy, may not be representative of all patients with epilepsy. These subjects also received therapeutic brain stimulation. We cannot exclude the possibility that stimulation influenced the rhythms we observed, but the stability of these rhythms despite parameter changes, including turning stimulation off, strongly argues against this. Given that patient subjective reports are notoriously inaccurate for seizure quantification[34], with systematic negative bias for certain seizure types (amnestic and nocturnal), we focused our study on objective quantification of electrographic seizures recorded with the device and not on clinical seizures. It is possible that clinical seizures have unique relationships to IEA rhythms[22], but our findings are consistent

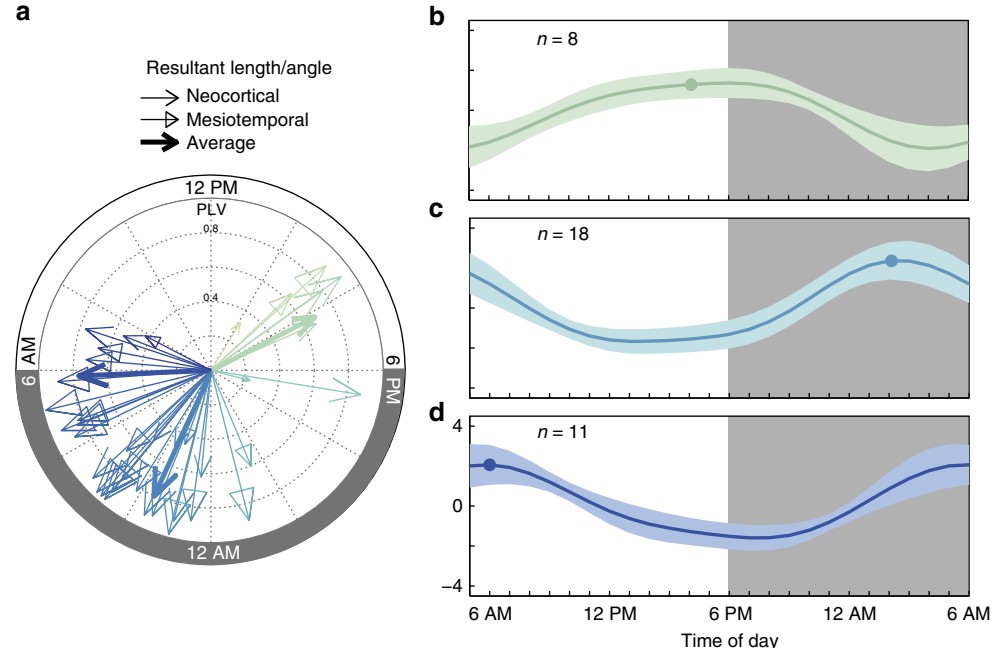

**Fig. 3** Circadian timing of peak IEA. **a** Phase entrainment of peak circadian rhythm to time of day for each subject ($N = 37$, 85–3478 days, resultant angle and phase-locking value (PLV), $p < 0.0001$ for all, Omnibus test, see Methods section) grouped into three clusters (group mean angle and PLV in bold, corresponding time as a dot in (**b**–**d**). Normalized average circadian amplitude ($\pm$SD) with peak in the late afternoon (**b**), early night (**c**), and early morning (**d**) were independent of seizure localization (mesial temporal vs. neocortical, $p = 0.14$, $\chi^2$-test) but may represent three chronotypes

with the nearly 80-year-old observation that clinical seizures demonstrate multidien periodicities in men and women[26]. IEA counts analyzed here depend on detection parameters that were dynamically adjusted based on clinical indications, and the RNS System stores limited continuous raw EEG. Changes in detection sensitivity impact the absolute IEA count, but our statistical approach accounts for this by relying on relative fluctuations within periods of constant detection settings. Finally, our study was retrospective, but leveraging knowledge of subject-specific multidien and circadian rhythms for prospective seizure prediction remains a major goal of future work.

Multidien rhythms have been identified in mood disorders[35], sleep patterns[30], and cardiovascular physiology[36], and their role in epilepsy will further fuel investigations into the underlying biological mechanisms. An endocrine basis seems likely[37,38]. For example, cortisol levels are positively correlated with IEA in some forms of epilepsy[39], and endogenous neurosteroids, hormonal modulators of GABA receptors, fluctuate over time and possess anticonvulsant properties[40]. Knowledge of these mechanisms and the ability to anticipate epochs of heightened seizure risk may enable dynamic, personalized treatment strategies[41].

## Methods

**Subjects**. We recruited 37 subjects (22 males) who had been implanted with the RNS System for purely clinical indications for at least 3 months and up to 9.9 years (median: 2.25 years) across two neurology centers (University of California, San Francisco, $N = 11$, and California Pacific Medical Center, $N = 26$). The two Institutional Review Boards approved the study and written informed consent was obtained from all subjects. Subjects had a variety of focal epilepsies (Supplementary Table 1). Indications for treatment with the RNS System as opposed to resective surgery included bilateral seizure localization (temporal and frontal), seizures arising from eloquent cortex (motor and visual), and previous contralateral resection.

**Data selection**. Detection of epileptiform activity by the RNS System relies on user-configurable tools (line length, area under the curve, and bandpass filtering) for which thresholds are optimized to detect seizure onset patterns[42]. Examples of subject-specific epileptiform activity detected by the RNS System have been reported previously[3] and are shown in Supplementary Fig. 1. Hourly detection

counts are stored by the RNS System for the last 28 days and are continuous as long as subjects download device data at least this often. For each subject, we discarded data recorded from the day of implant until reliable detection of seizures and IEA was achieved during the first few outpatient visits (median number of visits: 2, range: 1–11) a few months later (median: 29 days, range: 2–90 days; Supplementary Fig. 4). Sixteen subjects who did not download device data regularly had resultant gaps in detection counts. The data containing gaps longer than 6 days was considered discontinuous and analyzed in separate segments. The data of <90 days surrounded by gaps was discarded. Gaps up to 6 days were interpolated (see below). One subject (S4) underwent resective surgery and the device continued to record. This data were discarded. For visualization purposes daily counts were obtained by averaging 24 h of data on the same calendar date, but actual analyses were performed on the original hourly count data.

Due to memory constraints, the RNS System can store only a limited number of raw EEGs at a given time. Seizure detection with the RNS System relies instead on a surrogate marker, long-epileptiform activity (LEA; also termed 'Long Episode' by others[3,42]), which occurs when the EEG signal meets detection criteria for a clinician-specified length of time (typically, the minimum duration of each subject's electrographic seizure; LEA durations for this study ranged from 15 to 40 s, with an average of 26.8 s). Like detection counts, LEA timestamp information is stored by the device and available for analysis. Although generally a reliable proxy for seizures[43], LEA can also represent epochs of abundant IEA that do not meet criteria for electrographic seizures[44] (Supplementary Fig. 1b). To avoid contaminating seizure analysis with such false positives, we calculated for each subject the positive predictive value (PPV) of LEA for electrographic seizures (Supplementary Fig. 1c). For each subject, a Board-certified epileptologist (V.R.R.) assessed whether individual LEA corresponded to true electrographic seizures by visually reviewing corresponding EEGs. When possible (<300 LEA, 10 subjects), all individual EEGs were reviewed and each was labeled as an electrographic seizure (true positive) or as LEA other than an electrographic seizure (false positive). For subjects with too many LEA EEGs for comprehensive review ($N > 300$, 27 subjects), 100 EEGs were randomly selected from epochs with stable detection settings and a PPV was calculated for each epoch. Subjects ($N = 3$) who had <20 LEA were excluded from statistical analysis, because this was too low a number for a histogram-based statistical technique (see below). In total, we included 14 subjects for whom LEA was a reliable surrogate for seizures (PPV >90%; mean 98%, range 92–100%; Supplementary Table 1). Thus, we estimate that <2% of LEA used here may actually be sustained trains of IEA and that the rest represent true electrographic seizures.

**Time-series and wavelet analysis**. Hourly IEA counts were normalized ($z$-score) separately by block of recording between clinic visits so as to ensure stability of detection settings and anticonvulsant medications for each block. Continuous time-series were obtained by concatenating these individual blocks. Power and

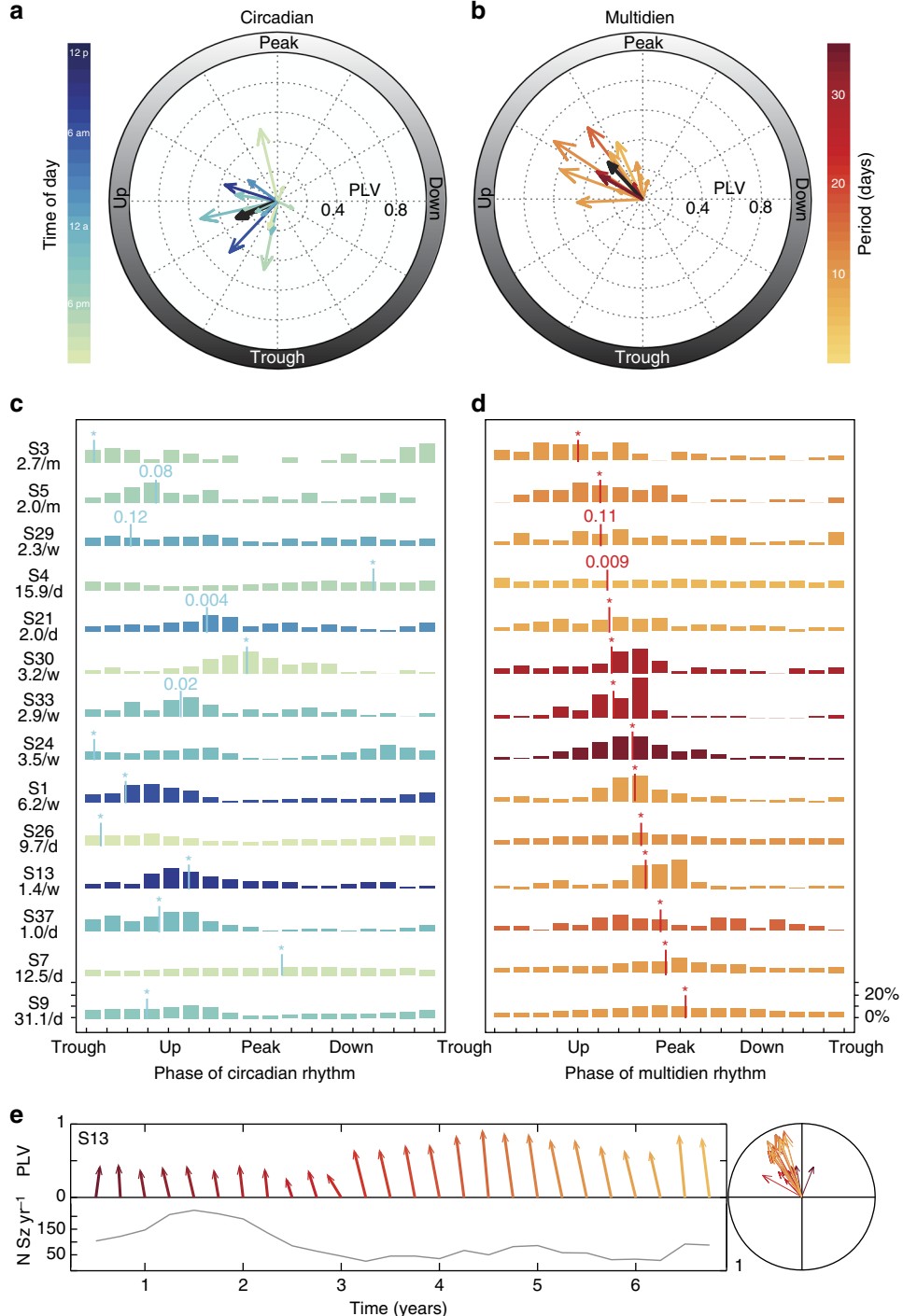

**Fig. 4** Phase preference of seizures in relation to underlying IEA rhythm. Seizure timing relative to phase of the underlying circadian (**a**) or multidien (**b**) rhythm for each subject shown as the PLV and resultant angle (*N* = 14; *p*-values in (**c**,**d**), Omnibus test, see Methods section). On average, the PLV was not different (*p* = 0.63, Wilcoxon test), but the angles were more tightly distributed and closer to the peak for the multidien as compared to the circadian rhythm (*p* = 0.002, Kuiper two-sample test, see Methods section). For visualization purposes, individual circular histograms of seizure counts (percentage of total count) for circadian (**c**) and multidien (**d**) rhythms are shown, ranked according to increasing multidien average phase (vertical bar position). *p*-values for the Omnibus test shown in Figure. \**p* < 0.001. Color codes in (**a**) and (**c**) are the same as in Fig. 3 and represent hour of the day of peak circadian rhythm. Color codes in (**b**) and (**d**) represent peak periodicity of multidien rhythm, illustrating that the preferred multidien phase is similar, regardless of the exact period length. **e** Feather and polar plots showing stable direction and magnitude of phase preference assessed every 3 months in one subject (S13). Color-coding establishes data correspondence between feather (left) and polar plot (right) and does not refer to color-bars in (**b**). The annual number of seizures is displayed to show the decrease over years of treatment with the RNS System and the number of seizures included in each calculation

phase of the hourly IEA counts time-series were obtained using a Morlet wavelet transform for 89 period bins (scales) with increasing spacing: 1.2 h between 2.4 and 31.2 h, 2.4 h between 33.6 and 48 h, 4.8 h between 2.2 and 4 d, 12 h between 4.5 and 10 d, and 24 h between 11 and 45 d. Gaps in recordings, if relatively short, were interpolated by a method similar to kriging in geostatistical and climatology research[45] (Supplementary Fig. 3). For each gap, the variance was calculated for

two windows, before and after the gap (each with same length as the gap itself). This variance was used to cast the normalization curves for pseudo-random data generated around a central tendency line directly connecting the means of the two peri-gap windows above. This process was performed for each gap, and only for gaps shorter than 20% of the period being analyzed (e.g., up to 2 days of interpolation allowed for a period of 10 days). Gaps were processed in increasing

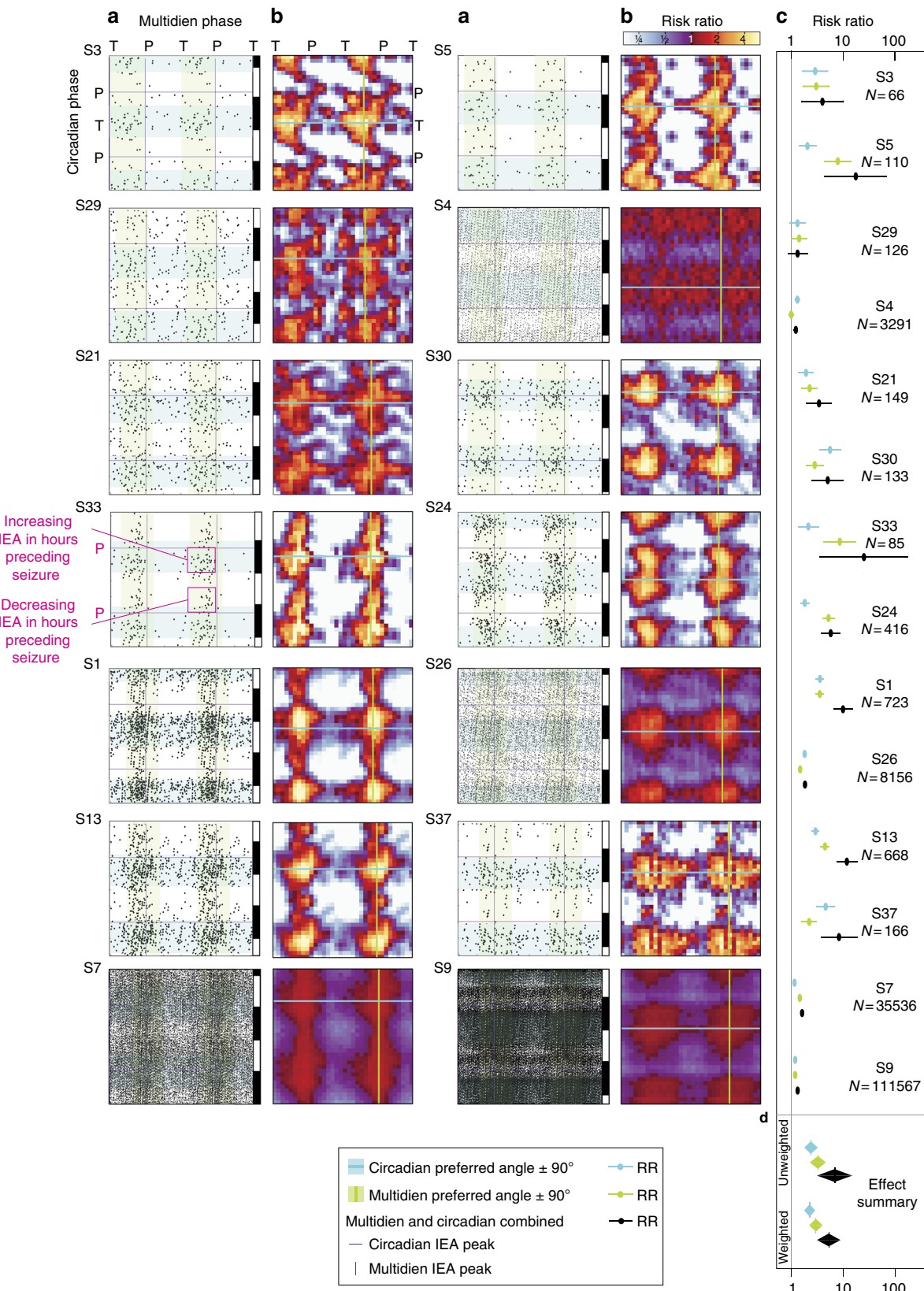

duration order, so that higher frequencies were not affected by interpolations performed at lower frequencies. In addition, a cone of influence—the region of the wavelet spectrum, shaped according to period length, in which edge effects impede accurate periodic estimation—was discarded at the extremities and around gaps too wide to be interpolated. The data could then be represented as a spectrogram of power or phase over time (time-frequency analysis) with excluded data at the extremities and around gaps (Supplementary Fig. 2). Power index in periodograms

was estimated for each individual scale (period bin) as the square root of the average over time of the absolute value of complex wavelet coefficients. Principal component analysis of all individual periodograms (Supplementary Fig. 3) was used to extract recurrent patterns. The six first components were selected as they explained 98.7% of the total variance in the frequency domain. Average periodograms were calculated for three separate clusters obtained by K-means (cosine distance, rank of 3) on the principal components coefficients (Fig. 2). Peaks in periodicity were defined as a positive-to-negative zero-crossing of the derivative of the periodogram.

**Instantaneous phase analysis.** Phase at every time-point was calculated in the frequency domain for a single band of wavelet coefficients corresponding to the peak period ±33.3% (e.g., 24 ± 8 h, or 15 ± 5 d), so as to accommodate variation in periodicity. Thus, at each time-point, the most powerful frequency within a range would most influence the phase. The average and variance in period length were calculated using the distance between two successive phase values of zero, excluding gaps. Circadian epileptiform peak activity histograms were obtained by counting occurrences within 1-h bins (24 bins, Fig. 3). Seizure phase histograms were obtained by counting occurrences within 20-degree phase bins (18 bins from $-\pi$ to $+\pi$) for circadian and multidien rhythms (Fig. 4). Similarly, average amplitude was obtained by calculating the mean and standard deviation of normalized data for all time-points with phase contained within these bins. If more than one multidien peak was present in the periodogram, we used the shortest (first) multidien periodicity to evaluate instantaneous phase. For visualization purposes, signal was reconstructed for the peak periods ±33.3% using an inverse wavelet transform (Fig. 1 and Supplementary Fig. 2). The phase-space representation allowed for the study of seizure risk as it relates to circadian vs. multidien phase. Scatterplots help visualize these relations abstracted from the fact that multidien periods vary across subjects.

**Statistics.** The sample size was determined by the availability of data. Previous studies have demonstrated quantitative analyses of these data sets with a relatively small number of subjects[5] (though our sample size is considerably larger in both number of subjects and recording durations). Circular statistics can be applied within-subject, and we investigated replicability across all subjects in our data set. The only pre-established exclusion criterion was length of continuous recording of <3 months, and we excluded seven recently implanted subjects for this reason. Given the observational nature of the study, there was no replication of measurements over time per se, though these long recordings and autocorrelation analyses (Supplementary Fig. 5) serve as technical measures of within-subject replicability. Values were expressed as mean ± standard deviation (SD) and plotted as dots and error-bars, unless specified otherwise. For categorical distributions (gender and seizure localization), we performed a $\chi^2$-test. For continuous variables, we performed a Wilcoxon test. Autocorrelations to the average periodogram were assessed at each time-point in the frequency domain using a Pearson coefficient and averaged over the total length of recording for each individual. For eight subjects with bilateral implants and >3 months of bilateral recordings, Pearson coefficients were calculated on the periodograms derived from each site. Other correlations were assessed using a linear model with intercept and the F-statistic vs. a constant model. Seizure rate was log-transformed for regression models, as it had a logarithmic distribution. Phase analyses were done using the 2012 Matlab circular statistics toolbox by Dr. Philipp Berens[46] including functions for circular mean, circular standard deviation, and resultant vector. We used the circular variant of k-means for clustering of angles. We used the Omnibus (or Hodges-Ajne) test to calculate statistical significance for non-uniform angular distribution (against the null hypothesis of a uniform distribution), as opposed to the more classical Rayleigh test, because some angular distributions were bimodal, especially in circadian rhythms. We used the Kuiper test (analog to the Kolmogorov–Smirnov test for circular data) to calculate the statistical difference between two angle populations,

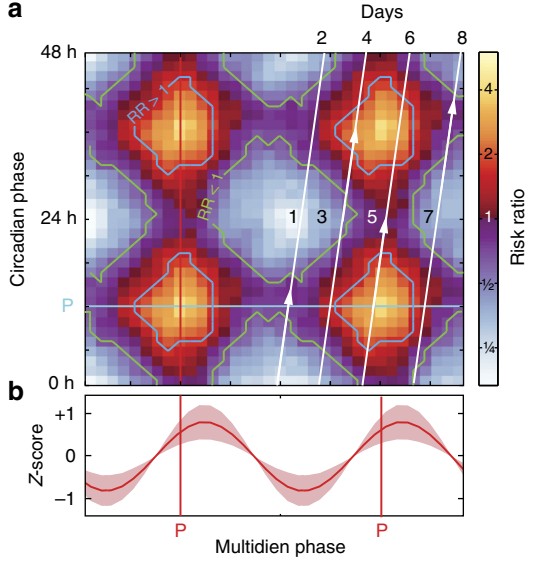

**Fig. 6** Average risk ratio (RR) map in the circadian vs. multidien phase-space. **a** Average of individual RR maps shown in Fig. 5 after alignment to the preferred phases ("*P*" in axes labels; red vertical line, multidien; cyan horizontal line, circadian). Blue and green contour lines indicate RR >1 and < 1, respectively, (95% CI excluding RR of one). To illustrate the concept of time-varying seizure risk, white lines depict the hypothetical circular trajectory of a subject with a 24 h circadian and 8-day multidien cycle. Each line covers two circadian cycles and a quarter multidien cycle. When starting on the left line, the subject mostly crosses areas of low seizure RR with the exception of medium RR at times of favored circadian timing (arrowhead). In the second quarter (second line from left), the subject crosses an area where multidien and circadian timing jointly increase seizure RR (arrowhead). In the third quarter (third line from left), the subject stays on an area of increased risk for two circadian cycles by traveling on a vertical band of favored multidien phase (arrowhead). The fourth quarter line joins the bottom of the first line to close the cycle. **b** Average multidien amplitude (z-scored) and peak position (just right of the preferred phase). Average circadian cycle is not displayed because the preferred phase was too variable across subjects, but circadian time is labeled 0–24–48 h

**Fig. 5** Individual risk ratio (RR) maps in the circadian vs. multidien phase-space. **a** Scatterplots of circadian vs. multidien phase at time of seizures (total number of seizures on the right of (**c**), *N*). Note that data have been duplicated on the x and y-axes to emphasize complete cycles. Each dot represents one to a few seizures happening during the same hour. *P*: peak, *T*: trough of underlying rhythms also represented with purple lines. The black (night, 6PM–6AM) and white (day, 6AM–6PM) boxes on the right y-axis represent approximate time of the day. Note the lack of correlation between the circadian and multidien angles (i.e., they do not align on a diagonal). Pink boxes in S33 highlight that, for a given multidien phase, IEA could go up or down in the hours before or after a seizure, depending on the circadian phase relative to peak (pink *P*). **b** Corresponding density plots representing risk ratio (color-coded logarithmic scale) for bins of 20-degree circadian and multidien phase combination. Each pixel represents the risk of having a seizure at this point in phase-space as compared to the risk of not having a seizure at this point. Green and cyan lines in (**b**) with corresponding green and cyan shading in (**a**) (±90°) represent preferred phases of seizures in relation to underlying IEA rhythms (also visible in Fig. 4). In some subjects, seizures can occur at any time of the day if in the at-risk multidien phase (S1, S5, S7, S13, and S24) and, conversely, in other subjects, seizures can occur on any day of the multidien cycle if at specific times of the day (S4, S26, and S37). **c** Forest plot showing the risk ratio for having a seizure when in-phase vs. anti-phase with the preferred phase of the underlying circadian or multidien rhythm or the combination of the two. **d** Effect summary for all 14 subjects. Overall, seizure occurrence was best explained by incorporating information about circadian and multidien rhythms

as it makes no assumption on the underlying distributions. These descriptive statistics, clustering methods, and tests in the circular domain have equivalents in the linear domain but they take into account the fact that angles flip from $\pi$ to $-\pi$, creating numerical discontinuity. The mean resultant vector is a metric of phase consistency of a population of angles. It is constructed by vectorial averaging of phase vectors each of unitary length equal to one. Thus, the angle of the mean resultant vector represents the mean angle of the population and its length (Phase-Locking Value, PLV) represents dispersion (values close to zero) or concentration (values close to one) of constituting angles. Given that the PLV is a continuous variable linearly varying from zero to one, we used a Wilcoxon test to compare population medians. To study the stability of this metric over time, we computed repeated PLVs every 3 months including data over a year ($\pm 6$ months at each time-point, Fig. 4f). To study the map of seizure risk for each individual subject, we binned seizure counts in the phase-space into 324 ($18 \times 18$) 20-degrees circadian and multidien phase combination elements. To reduce noise, we smoothed this result with a 2D Gaussian kernel spanning two standard-deviations over $\pm 40$ degrees. Each single bin, containing the number of seizures that occurred at a given phase combination, successively represented true positives (TP), whereas seizure counts in the 323 other bins represented false negatives (FN). False positive (FP) were the number of occurrence of a given phase combination (without seizures), and True Negative (TN) were all the other phase combinations (without seizures). The risk ratio was calculated as $\frac{TP/(TP+FP)}{FN/(TN+FN)}$. This was done for each bin resulting in a map of 'risk factor-seizure' association (Fig. 5). Thus, the risk ratio takes into account the number of seizures observed at a given phase combination, the seizure rate of a given subject, and the probability of finding a given combination of phases. These maps were averaged across subjects after alignment to the preferred circadian and multidien phases (Fig. 6). In addition to this visualization of risk-ratio maps, we estimated the global effect size with confidence intervals for each individual, by grouping bins of the map that were in-phase (the 'risk factor') or anti-phase with the preferred angle ($\pm 90°$) and calculating a risk-ratio between these two categories (Fig. 5). This was done for circadian and multidien rhythms independently and for the combination of the two phases (using a union Boolean operator). Finally, we displayed risk ratios as a Forest plot and computed an effect size summary metric across subjects using two methods: (i) a simple unweighted average of the risk ratio and standard error, and (ii) a random effect weighted average of the risk ratio and standard error (variance in risk-ratio was assumed to be non-random, Fig. 5)[47].

**Data availability**. The data and code utilized in this study are available from the corresponding author upon reasonable request.

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

## Acknowledgements

This work was supported by the National Institutes of Health (NIH) grants R25NS070680-07 (J.K.K.) and R01-DC012379 (E.F.C.). The authors are grateful to Phil Thornton for technical assistance during data collection.

## Author contributions

M.O.B. and V.R.R. designed and initiated the study. V.R.R., D.K.-S., and E.F.C. recruited subjects from their clinical practices. V.R.R., E.A.M., and J.C.A. performed the data collection. M.O.B. and J.K.K. performed the data analyses. V.R.R., M.O.B., and J.K.K. wrote the manuscript.

## Additional information

**Competing interests:** V.R.R. and D.K.-S. have received honoraria from NeuroPace, Inc. for consulting and speaking engagements. E.A.M. is an employee of NeuroPace, Inc. M.O.B. is a part-time employee of the Wyss Center for Bio and Neuroengineering. The authors declare no targeted funding or compensation from NeuroPace, Inc. for this study. The remaining authors declare no competing financial interests.

