## [Peer review File · Nature Communications]

Reviewers' comments:

Reviewer #1 (Remarks to the Author):

This is an interesting paper regarding the relationship between interictal and ictal activity in human patients recorded from chronically implanted electrodes on the cortical surface. The authors analyze interictal spike patterns in 37 patients and seizures in a subset of 14 patients, and present two major findings: first, that interictal spike frequencies demonstrate periodicities at both circadian and multi-day periods; and second, seizures occur during a multi-day increase in the frequency of interictal spikes. Comments:

1. The first finding, that there are repeatable multi-day periodicities to interictal spike frequencies, is interesting and should be presented in more detail. For example, the periods should be displayed as a graphic, with gender as a displayed variable (see point 3). Figure 2 presents periodicity analyses that appear to be smoothed across the 37 subjects; a minor point is that the shading is not explained.

1A. The repeatability of the spike frequency period is an important parameter. Clinically, patients often have cyclical seizures but the period varies so much that it is rarely predictive. What does the interictal spike autocorrelation function look like for each of the patients plotted in supplemental figure 1? For some patients there may be a clear periodicity e.g. the individual represented in figures 1f, 2a and supplemental figure 2a,b; while for many others in supplemental figure 1, there does not appear to be a sustained periodicity.

2. In acute recordings in epilepsy monitoring units, the interictal spike rate was observed to increase for up to several days after seizures (Gotman and Marciani, Ann Neurol 1985). These patients were being withdrawn from anticonvulsants, so the data needed confirmation. Here, Figure 4b demonstrates that seizures occur neither at the peak nor the trough of the interictal spike frequency. Rather, the spike frequency increases after seizures. This postictal increase in spike rate is also evident in figure 1f. The postictal increase in spike rate is very similar to what was shown in the earlier publications of EMU data. The new advance here is that the patients are at steady-state, i.e. not being withdrawn from anticonvulsants.

3. It is not clear to what extent the multi-day periodicities described in lines 61-63 are catamenial. How many of the patients experiencing ~ monthly periods were women? How many of the 14 patients with phase-locked interictal and ictal activities were women? How many of the phase-locked periods could be explained by catamenial cycling?

4. Only a minority of patients (14) were included in the ictal-interictal pattern matching. This is troubling; the logic for many of the exclusions is not clear i.e. patients with < 20 seizures. This needs a lot more explanation in the body of the text. In chronic recordings, interictal spikes are much more difficult to identify and separate from artifact than seizures, because they are so transient – there is no pattern of activity that can be recognized. Thus it is not clear why seizures could only be reliably identified in 14 of 37 patients.

5. The interictal and ictal data were recorded from a stimulating device. Was this a closed or open loop stimulator? To what degree could the pattern of stimulation have affected the results?

6. Minor – the interictal spike illustrated in figure 1a looks like an artifact. What are the time bases for the recordings in figures 1a and 1b?

Reviewer #2 (Remarks to the Author):

The main claim of the paper is that seizures are locked to multidien rhythms in epilepsy. The paper is essentially descriptive, but sometimes initial observations are more important than mechanisms. Finding an ultraslow modulation of seizures is very important for the epilepsy field. However, in its present state, the work is not convincing. If the authors can provide better convincing arguments, I would be happy to fully support this potentially landmark paper. Although I like the concept, I am not convinced by the analysis that the authors performed,

because they did not provide enough details, and/or because they are using statistics I am not familiar with.

I'll list a few of my concerns regarding statistics, but I would feel better if the authors could have their stats checked by a highly versed person.

Major.

1. The way interictal spikes are detected is not described. Yet, this is the foundation of the paper. There are many types of spikes and several groups (Kullmann, Stead-Litt-Worrell) have designed semi-automated procedures to detect and classify interictal activity. I don't understand why the method section mentions Long Epileptiform Activity. Did the authors only include bursts of spikes (versus isolated)? What do you make of the time interval between two LEAs? Also, Chauviere (2012) reported LEA dynamics before the first spontaneous seizures. Do you see different types of spikes in LEAs? Avoli's work may be checked. This would require more complicated signal analysis (perhaps for another paper - or it could reinforce the present story - I don't know).

2. How were seizures detected? This is not described. The seminal Cook paper demonstrates that most seizures are subclinical (at least the patients are not reporting them). Could it be that only the clinical seizures are phase locked, and the subclinical are dispersed in other phases?

3. Stats and analyses appear too complicated, and sometimes do not have any meaning (to me, and to internet). What is the autocorrelation Pearson coeff? I am not convinced by the slow modulation of LEAs. It is very clear for the example shown in fig 1, but when we look at supp fig 1 of all patients, we fail to see any peak after 24h in the periodogram in many patients with seizures (e.g. S7, S9).

I don't understand how the slow frequency of interictal activity was calculated. From the graphs, it seems very variable within patients. Why do you want to force an oscillation in this slow rhythm? Would it be more simple to look at interictal spike dynamics (acceleration?). I am not convinced that the authors could extract a rhythm in all patients (except fig 1). You need a much better detailed description and validation of what you did. The cluster analysis does not make sense to me. You should do it on an individual basis and not lump the results. Each patient appears unique based on fig 4.

In fig 4, you should calculate the stat significance for each bin.

Also, you need to use surrogate analysis to demonstrate significance.

You tested H0, but I would test H1 as well.

All considered, the title is misleading, you do not show phase locking, you show phase modulation, as the majority of seizures are outside the bins of the preferred phases in many patients.

Phase locking has no strict definition in terms of how much of the events must be in a specific phase, but for place cells, the firing phase is nicely predictive of the spatial position of the animal.

4. Was neurostimulation always on and continuous? If not, you need to re-analyze the data to take into account this factor.

In summary, I think that the paper could influence thinking in the field of epilepsy. However, the authors failed to convince me. I asked statisticians of my team, and they were puzzled by the unnecessary complexity of the approach (they were not convinced either). I would be delighted to be convinced. I recommend giving the authors another chance.

Reviewer #3 (Remarks to the Author):

This ms from Baud and colleagues describes long-term recordings from epileptic patients which suggest that the frequency of intracranially recorded interictal events varies with a multi-day as well as an intra-day rhythmicity. Furthermore, in recorded patients, the occurrence of ictal events seems to be linked to a specific phase of the multi-day rhythms. These hypotheses are quite well supported by the data presented, even if the wavelet fit to time sequences is not so convincing. It is a shame that the study did not include a predictive element. The data should be of interest to the wider community and poses several interesting questions for future work. What is the basis for the slow rhythm and how does it influence seizure generation?

Specific points:

1. The slow rhythm shown in Fig. 1d does not really have a sinusoidal aspect. Superimposed on the diurnal changes, this data seems rather to show a relatively sudden (2-4 days) increase in the frequency of hourly counts which is followed by a much slower decay (10-15 days). So how exactly was the orange multidien curve derived? Was this pattern typical for all the patients? How could the conclusions change with a more realistic curve fitting?
2. In the year-long recording of Fig. 1f, there seem to be some amplitude constraint on the orange curve, such that it misses the peaks of the highest daily counts (April-June). What is it? Was there a similar variability in amplitude of multi-day peaks in interictal frequency for other patients?
3. The phase window for seizure occurrence (Fig. 1g) depends on the orange curve. How does interpretation change if the orange curve does not reflect the data faithfully. In Dec 2015 and July 2016 for instance (Fig. 1f) multi-peaked changes in count number are reduced to a single sinusoid.
4. In a subset of women, seizures occur with frequency linked to the menstrual cycle. The ms notes carefully that men and women both exhibited multi-day periodicity in interictal event occurrence. Still it would be interesting to know if the study picked up any of this group of women with epilepsy.
5. Fig. 2. The parameter on the vertical axis might be better described. Amplitude of what?
6. The data of Fig. 4b seems to show a phase specificity for the association between seizure occurrence and slow variations in counts of interictal events. As in Fig. 1 though it seems to be based on a poor sinusoid-like wavelet fit to changes in frequency of interictal events. If seizure occurrence depends on some state parameter associated with increasing interictal frequency, why not plot seizure occurrence against day-to-day changes in interictal event counts? The author's hypothesis posits that seizures are closely associated with day-to-day increases in interictal frequency.
7. On the possibility of prediction: Maybe as a next step, it could be informative to define a long-term seizure rhythm from say 12 months data, and then test the accuracy of predicted dates of seizure occurrence against real seizure events over the next 6 months. If the window of peak seizure susceptibility is 2-4 days per month, as this data suggests, verification of the accuracy of predictions might open the way to time-limited protective measures or treatments. Some variant of this approach might be discussed.

Reviewer #4 (Remarks to the Author):

This paper is a robust analysis of data from the RNS over many months in several patients, and shows how detected epileptiform events have periodicity on multiple time scales, and seizures are more likely during these periods. This is a very unique dataset that provides information about the long term seizure risk. The authors show not only that most patients (in this small dataset) have diurnal and multidien variation in the IEA, but also that seizures were more likely to occur at specific intervals in these variations. This information is of interest to the larger community, helps clinicians and scientists understand the long term variability of seizures, and helps answer a longstanding debate about the utility of interictal discharges. The statistical analysis is advanced, and an appropriate method to deal with these cyclical statistics (although they are likely unfamiliar to most readers). There are some concerns with the paper.

Storage limitations One of the primary limitations of this RNS device is its very limited storage, and the method of saving data. The device only saves events that are designated for detection, and it only can save a limited number of those before starting to overflow and dropping earlier detections. The patients have to upload data regularly to assure detections are not lost. Thus this is a very limited dataset to begin with (only saving clips of detected events), with serious concerns about lost data (from when there is overflow or longer periods without an upload). These limitations are addressed to some degree in the paper, but they are not really explained. Thus this

is not “10 years” of data, but 10 years in which a very limited subset of detected events may or may not have been uploaded. This leads to a couple of questions and a concern. How do the authors know there wasn't lost data from overflow? How can they be certain there weren't gaps in uploading (i.e. did patients really upload data daily for 10 years? That seems remarkable.) How can they be certain the uploading contained all data since the last upload? The paper should make it very clear that these limitations exist. (More questions on gaps below) The RNS only records what you tell it to record. This is important, because the only other device with similar data (the Neurovista device) recorded ALL of the data, and thus this dataset is at a clear disadvantage. This limitation needs to be very clearly stated.

Algorithm limitations That brings up another concern with what actually gets recorded. What is the sensitivity/specificity of these detections? (in these patients, I suspect it will not be possible to test this). There is one surrogate for specificity--how often were there magnet swipes/reported symptoms that were NOT detected by the algorithm? How were these “false negatives” dealt with? My guess is that algorithms were dynamically adjusted to improve sensitivity. But then, if the algorithms were changed to detect different events, how do you reconcile all the previously-undetected events that were not detected? The tone of this article implies an unchanging, infallible detector in each patient, which is really what is needed to make this study ideal. If the algorithms changed, this will need to be clearly stated and reconciled. And the paper should make it very clear what the limitations of the algorithm are.

The algorithm and RNS system also have their own unique nomenclature, which are not well explained. An “IEA” in the RNS can mean several things, but often is detecting abnormal discharges or runs of activity, rather than individual spike and waves. Most readers, however, are going to assume “IEA” refers to basic spikes—is that so? Are the authors claiming that they have detected all spikes? If not this needs to be clearly stated (the paper really does imply that that is what is happening, especially when bringing up the controversy of spike rate and seizure risk). There needs to be a clear section defining how IEA are chosen, what typical IEA entailed, and what types of IEA might not be detectable with RNS, how many spikewaves are/are not detected, etc. Figure 1 seems to imply it is detecting spikewaves, but in this reviewer's experience with RNS that is not what is typically detected.

Use of the word “random” One of the main results of the paper is that seizures are more or less likely at different times: Intro: “Our findings indicate that seizures are not random events and that slow rhythms of IEA are a critical biomarker for seizure prediction.” Also p. 6 “seizures are non-random events organized by underlying biological rhythms operating over long timescales.” These statements are not exactly true and should be said with more care. Seizures are in fact still ‘random’—on any given time scale they are not deterministic, and there is no time when they can be predicted perfectly. What is happening is that there are times when they are more likely, and it would be easier to predict. What the authors are showing is that the underlying RISK is variable, with individual seizures still occurring at random times (albeit affected by a variable risk), very similar to past modeling work shown in Jirsa et al Brain 2014 and several articles by Lopes da Silva. Also the phrase “biomarker for seizure prediction” is too ambiguous. It is really a biomarker of seizure risk, which then can be used to inform seizure prediction.

Past work In addition to the modeling work above, there have been other physiological studies showing how EEG can change over longer time periods to suggest ictogenicity. For instance, Huberfeld et al. Nat Neurosci 2011, Karafin et al Seizure 2010, Chauviere et al Ann Neurol 2012).

Proving hypotheses. The statement that “For example, seizures preferentially occur during the rising phase of a multidienn IEA cycle, but this could coincide with the peak or the trough of the circadian IEA cycle, as observed by others, and shorter timescale studies would draw seemingly contradictory conclusions” is very intriguing. However, could the authors please show if and when that happens? I don't know of another time when similar data will be presented—this paper has the data ready right now! Please inform us about whether the ultradian and circadian phases actually have such a phase mismatch, and how often it occurs. If it does not happen in your data,

then this statement is probably incorrect.

In addition, the authors talk about how this would help with prediction. This is never proven and there is never any attempt to do so. It is probably outside the realm of this paper, but care should be taken with such statements—they have not proven that it will have any bearing on seizure prediction.

Gap filling is very important with this device, and is a clever way to deal with potential problems in the data. But some things are still not clear. How can we be sure whether events were discarded? What type of spline was used for interpolation; was it always monotonic? What was the “cone of influence”? That was not clear. Furthermore, looking at the data it is not clear how these tools are being used. For instance in sup Table1, it says patient S13 has 73% valid data, but looking at Supp Fig 1 it is not clear where the gaps are (and in that figure I cannot make out where any of the “interpolated” purple segments are).

Discernability of differences. Fig 4 shows the actual differences in phase, but I am not sure all of those patients had a clear difference. Why is the amplitude of the difference smaller with larger N in the multidien (S4, 26, 7, 9 are all the highest N, and all have the “flattest” distributions)? That is concerning—the difference appears trivial in all the patients with the highest N (which is usually 1-2 orders of magnitude higher than the others). With such high N, “statistical significance” is somewhat guaranteed, but is there actually a discernable difference? I doubt that multidien variations would be helpful in those patients, as there just isn’t enough of a difference to measure. And if these distributions tend to “flatten” with longer recordings, will discernability vanish for all patients? This is actually probably the most important concern in the paper.

Also, there is no legend for the X axis in that figure.

Reviewer # 1:

This is an interesting paper regarding the relationship between interictal and ictal activity in human patients recorded from chronically implanted electrodes on the cortical surface. The authors analyze interictal spike patterns in 37 patients and seizures in a subset of 14 patients, and present two major findings: first, that interictal spike frequencies demonstrate periodicities at both circadian and multi-day periods; and second, seizures occur during a multi-day increase in the frequency of interictal spikes.

Comments:

1. The first finding, that there are repeatable multi-day periodicities to interictal spike frequencies, is interesting and should be presented in more detail. For example, the periods should be displayed as a graphic, with gender as a displayed variable (see point 3).

We have added to Figure 2 a new panel (c) with histograms, stratified by gender, showing the number of subjects who had IEA periods of various lengths. The distributions of multidien periods are similar ($p=0.87$, chi-square test) for male and female subjects.

Figure 2 presents periodicity analyses that appear to be smoothed across the 37 subjects; a minor point is that the shading is not explained.

Fig. 2b shows average periodograms for subjects within a cluster. We also show individual periodograms elsewhere (Supp. Fig. 4). The shading in Fig. 2b represents ± 1 SD. This information has been added to the Figure Legend.

1A. The repeatability of the spike frequency period is an important parameter. Clinically, patients often have cyclical seizures but the period varies so much that it is rarely predictive. What does the interictal spike autocorrelation function look like for each of the patients plotted in supplemental figure 1? For some patients there may be a clear periodicity e.g. the individual represented in figures 1f, 2a and supplemental figure 2a,b; while for many others in supplemental figure 1, there does not appear to be a sustained periodicity.

We agree that the stability of IEA rhythms is critical, and we acknowledge that some subjects appear to have a more discrete multidien peak in the periodogram than others. We directly address the issue of period stability in Supp. Fig. 5e (previously Supp. Fig 2e), which shows that the periodogram autocorrelation for all 37 subjects is fairly robust (0.72 ± 0.07) even over long periods of time. However, there is some variability in period length within subjects and this has the effect of flattening the multidien peaks in periodograms averaged over long recording durations. Importantly, we factored in multidien period variability in the seizure timing analysis by computing the instantaneous phase (red dotted line in new Supp. Fig 2). Supp. Fig. 4 (previously Supp. Fig. 1) now shows that multidien peaks tend to be sharper in periodograms averaged over shorter timescales (blue lines). Despite variability in multidien period length,

seizures can remain tightly coupled to specific phases of this rhythm over many years (for example, see feather plot in new Fig. 4e).

2. In acute recordings in epilepsy monitoring units, the interictal spike rate was observed to increase for up to several days after seizures (Gotman and Marciani, Ann Neurol 1985). These patients were being withdrawn from anticonvulsants, so the data needed confirmation. Here, Figure 4b demonstrates that seizures occur neither at the peak nor the trough of the interictal spike frequency. Rather, the spike frequency increases after seizures. This postictal increase in spike rate is also evident in figure 1f. The postictal increase in spike rate is very similar to what was shown in the earlier publications of EMU data. The new advance here is that the patients are at steady-state, i.e. not being withdrawn from anticonvulsants.

We agree that our data confirm and extend prior observations on the relationship between IEA and seizures. We have revised the Discussion by including reference to the work of Gotman and Marciani and by highlighting that a strength of our study is the use of chronic recordings in subjects who remain on anticonvulsants. However, some studies have shown that increased post-ictal spiking is not observed in all cases^{1,2}, and we believe that this apparent discrepancy results from the dynamic interplay between circadian and multidien rhythms. We now include a new analysis (Supp. Fig. 8) showing that, for seizures occurring at a given circadian phase, IEA trends before and after the seizure are determined by the corresponding multidien phases, and vice versa.

3. It is not clear to what extent the multi-day periodicities described in lines 61-63 are catamenial. How many of the patients experiencing ~ monthly periods were women? How many of the 14 patients with phase-locked interictal and ictal activities were women? How many of the phase-locked periods could be explained by catamenial cycling?

Nearly all of the 37 subjects (22 males, 15 females) showed multidien periodicities (Supp. Fig. 4), and the distributions of periods were similar in male and female subjects (Fig. 2c). Of the subset of 14 subjects meeting criteria for seizure analysis, 9 (64%) were male and 5 (36%) were female. Of the 5 female subjects included in the seizure analysis, one (Subject S29) had catamenial epilepsy based on clinical reports, and she was found to have IEA periods at 13 and 26 days, which could be consistent with catamenial cycling. Overall, though, 14 out of 15 females in the study had no clinical evidence of catamenial seizure variation. Thus, while we cannot exclude a contribution of catamenial cycling, it seems unlikely to explain fully our observations about seizure timing. We now include these points in the text, and we indicate in Supp. Fig. 4 which female subject (labeled '(c)') was thought to have catamenial epilepsy on clinical grounds.

4. Only a minority of patients (14) were included in the ictal-interictal pattern matching. This is troubling; the logic for many of the exclusions is not clear i.e. patients with < 20 seizures. This needs a lot more explanation in the body of the text.

We apologize for the lack of clarity. Due to memory constraints, the RNS System neurostimulator can store only a limited number of ECoGs, so visual confirmation of every electrographic seizure is not possible. Instead, seizure detection with the RNS System is based on a count system, relying on a surrogate marker called Long Epileptiform Activity (LEA; also termed ‘Long Episode’^{3,4}), which identifies instances in which detection of epileptiform activity persists beyond a clinician-specified length of time (typically, the minimum duration of each subject’s electrographic seizure). Unlike ECoGs, the dates/times of LEA are not overwritten or lost so long as subjects download device data at least once every 28 days. To minimize false positives (i.e. LEA that are not electrographic seizures), we calculated for each subject the positive predictive value (PPV) of LEA for electrographic seizures by systematically reviewing LEA in stored ECoGs. Subjects were included in seizure analysis if PPV exceeded 90%, which is higher than the cutoff (75%) in a previous study of seizure timing using RNS System recordings³. The stringency of our criteria allowed inclusion of a minority of subjects in the seizure analysis, as pointed out by the Reviewer, and we certainly could have opted for a lower threshold. However, we feel that our conservative criteria and the high fidelity of seizure detection in the included individuals lends credence to our results.

To yield robust results, circular statistics require enough observations. We set the minimum at 20 seizures based on a goal of having approximately 1-2 observations per histogram bin (18 bins of 20 degrees each) to rule out a uniform distribution. Excluded subjects tended to be those with shorter recording durations. To clarify these points, the Methods section has been substantially revised and a new Supp. Fig. 1 with illustrative ECoGs has been added. Also, please see our response to the next point.

In chronic recordings, interictal spikes are much more difficult to identify and separate from artifact than seizures, because they are so transient – there is no pattern of activity that can be recognized. Thus it is not clear why seizures could only be reliably identified in 14 of 37 patients.

The RNS System detects seizures and IEA in real-time but cannot store continuous ECoG, so nearly all of the data we utilize in this study are the counts-per-hour of these detections. The detection algorithms implemented in the RNS System neurostimulator are designed to be computationally efficient for real-time detection, and they rely on three highly-configurable tools (line length, area under the curve, and bandpass) for detection of epileptiform activity⁴. LEA frequently, but not always, represent electrographic seizures, and they are marked when ECoG signals exceed detection thresholds in a sustained manner. In patients with frequent interictal discharges, for example, the detection tools are sometimes too limited to distinguish electrographic seizures from periods of abundant spiking, hence the rationale for our PPV criteria described above. This point is now illustrated in a new Supp. Fig. 1, but, for convenience, an example is also provided below:

Subject S21

This ECoG shows epileptiform activity meeting detection criteria based on bandpass tool for longer than 30 seconds (only 10 sec shown here). The RNS System therefore recorded the event as LEA, but visual inspection reveals frequent spiking but not an electrographic seizure:

An ECOG from the same subject on the following day shows another LEA (i.e. a detection event sustained for longer than 30 sec; again, only 10 sec shown here) which is clearly an electrographic seizure:

Thus, rather than assuming all LEA are electrographic seizures, we calculated a PPV for each subject by reviewing available ECOGs and found, using stringent criteria, that LEA was a highly reliable proxy for seizures in 14 out of 37 subjects. Seizure timing analyses were performed only in these subjects.

5. The interictal and ictal data were recorded from a stimulating device. Was this a closed or open loop stimulator?

The RNS System is a closed-loop neurostimulation device⁵. We have added this important clarification to the Introduction.

To what degree could the pattern of stimulation have affected the results?

As we acknowledge in the Discussion, we cannot rule out the possibility that stimulation affected our results in some way because all subjects received stimulation for clinical purposes. However, we are doubtful that responsive stimulation is driving the rhythms we observe for several reasons:

- (1) Subjects differed widely with regard to RNS System lead locations (Supp. Table 1) and ‘stimulation pathway’ (i.e. designation of which electrode contacts serve as anodes and cathodes during current delivery), making it unlikely that potential effects of stimulation on IEA rhythms and seizures were consistent enough to explain our results.
- (2) In clinical practice, stimulation parameters (e.g. frequency, intensity, burst duration, pulse width) are typically adjusted at least every 2-3 months for the first 1-2 years post-implant and are thus far more dynamic than the observed variability in multidien period length within subjects (Supp. Fig. 5e), suggesting that stimulation and multidien rhythms are not tightly coupled.
- (3) In previous studies, weeks-long cycles of spike rates were found in some subjects with an implanted seizure advisory system that records chronic intracranial EEG but does not deliver brain stimulation^{6,7}.
- (4) We now include in Supp. Fig. 4 (previously Supp. Fig. 1) an illustration of multidien rhythms during epochs when the stimulation function of the device was turned OFF in a few subjects (green lines). IEA rhythms continue during these periods, strongly arguing that stimulation is not directly responsible for these rhythms.

We have now expanded discussion of the possible influence of stimulation on our results in the manuscript.

6. Minor – the interictal spike illustrated in figure 1a looks like an artifact. What are the time bases for the recordings in figures 1a and 1b?

The ECoG timebase (20 sec) is now shown more clearly in Fig. 1b, c, and we have added an inset in Fig. 1c that more clearly shows the epileptiform spike morphology. Additional examples of epileptiform activity detected by the RNS System are also shown in the new Supp. Fig. 1.

Reviewer #2:

The main claim of the paper is that seizures are locked to multidien rhythms in epilepsy. The paper is essentially descriptive, but sometimes initial observations are more important than mechanisms. Finding an ultraslow modulation of seizures is very important for the epilepsy field. However, in its present state, the work is not convincing. If the authors can provide better convincing arguments, I would be happy to fully support this potentially landmark paper. Although I like the concept, I am not convinced by the analysis that the authors performed,

because they did not provide enough details, and/or because they are using statistics I am not familiar with. I'll list a few of my concerns regarding statistics, but I would feel better if the authors could have their stats checked by a highly versed person.

We agree that our analytical methods are complex, and we understand the importance of making our approach accessible to a wide readership. Still, we believe our methods are appropriate for the questions being asked and for the nature of the data. Indeed, Reviewer 4, whose comments were solicited by the Editor specifically to evaluate the suitability of the mathematical methods, agrees, stating, *“The statistical analysis is advanced, and an appropriate method to deal with these cyclical statistics”*. We have now added more details to the manuscript and substantial supplementary material (for example, see new Supplementary Figures 1 and 2) to clarify our methodology for a wide audience.

Major.

1. The way interictal spikes are detected is not described. Yet, this is the foundation of the paper. There are many types of spikes and several groups (Kullmann, Stead-Litt-Worrell) have designed semi-automated procedures to detect and classify interictal activity.

We apologize for any confusion on this critical point. Detection of interictal epileptiform activity (IEA) by the RNS System is detailed in a response to Reviewer 1 above, and the Methods section has now been substantially revised for increased clarity. As previously described by Spencer and colleagues³, IEA detection parameters were implemented and refined by clinicians through an iterative process of review and reprogramming in routine outpatient visits to arrive at optimal detection parameters for each subject. Using three customizable tools (line length, area under the curve, and bandpass), detection parameters were typically configured to detect epileptiform activity present at the electrographic onset of a seizure. Although the detection parameters were not uniform across subjects, detected activity generally fell into one of three categories: spiking, rhythmic alpha/beta, and low-voltage fast activity. We now include a new Supp. Fig. 1 illustrating the types of epileptiform activity detected by the RNS System.

I don't understand why the method section mentions Long Epileptiform Activity. Did the authors only include bursts of spikes (versus isolated)? What do you make of the time interval between two LEAs? Also, Chauviere (2012) reported LEA dynamics before the first spontaneous seizures. Do you see different types of spikes in LEAs? Avoli's work may be checked. This would require more complicated signal analysis (perhaps for another paper - or it could reinforce the present story - I don't know).

As described in a response to Reviewer 1, Long Epileptiform Activity (LEA) refers to detection of sustained epileptiform activity persisting beyond a clinician-defined length of time such that it can serve as a surrogate for electrographic seizure activity. However, since LEA occasionally represents periods of abundant interictal spiking that do not qualify as electrographic seizures (see example ECoGs above), similar to the interictal-like activity (“ILA”) studied by Chauviere and colleagues in rats⁸, we took great care to determine the validity of LEA as a proxy for seizures in each subject. Thus, analysis of seizure timing was only performed on those subjects

for whom the Positive Predictive Value of LEA for electrographic seizures exceeded 90% (mean 98%). For clarity, this concept is now illustrated in a new Supp. Fig. 1.

2. How were seizures detected? This is not described.

Please see our responses to the previous comment and to Reviewer 1. Dates/times of LEA are logged by the RNS System and were used as a proxy for electrographic seizures in subjects for whom this was determined to be a valid assumption (PPV > 90%).

The seminal Cook paper demonstrates that most seizures are subclinical (at least the patients are not reporting them). Could it be that only the clinical seizures are phase locked, and the subclinical are dispersed in other phases?

It is certainly possible that clinical and subclinical seizures have different relationships to underlying rhythms⁹, and we have added this interesting point to the Discussion. However, given that patient subjective reports are notoriously inaccurate for seizure quantification¹⁰, and that reliance on clinical seizure counts would likely bias away from some seizure types (dyscognitive/amnestic) and certain times of day (nocturnal seizures), we elected to study only electrographic seizures. In clinical experience with the RNS System¹¹, most seizures are subclinical, consistent with the study from Cook and colleagues⁶. Interestingly, a (historic) 1938 paper by Griffiths and Fox examined counts of clinical seizures over unprecedented timescales (years at a live-in colony for patients with epilepsy) and observed in men and women multidien periodicities similar to our electrographic findings. We have cited these papers and included the points above in the manuscript text.

3. Stats and analyses appear too complicated, and sometimes do not have any meaning (to me, and to internet). What is the autocorrelation Pearson coeff?

We acknowledge and regret that the statistical methods used here may not be familiar to all readers, yet they are the most appropriate and rigorous way to analyze cyclical time-series data. We have reworded multiple sections of the text to increase accessibility for a wider audience. Wavelet transforms and related statistics are widely accepted techniques for time series analyses in neuroscience¹²⁻¹⁷ and other diverse fields, such as geophysics¹⁸, epidemiology¹⁹, economics²⁰, and ecology²¹. We have added a new Supp. Fig. 2 that illustrates wavelet decomposition in a step-by-step manner. Phase analyses were performed using a published Matlab toolbox (CircStat).

Pearson coefficients are correlation statistics used to characterize the relationship between two datasets, ranging between -1 (perfect negative correlation) and +1 (perfect positive correlation). In Supp. Fig. 5b (previously Supp. Fig. 2b), we show Pearson coefficients between the average periodogram and its own component periodograms across different time windows over the entire recording period, thus representing an “autocorrelation” function.

I am not convinced by the slow modulation of LEAs. It is very clear for the example shown in fig 1, but when we look at supp fig 1 of all patients, we fail to see any peak after 24h in the periodogram in many patients with seizures (e.g. S7, S9).

Please see our response above to a similar concern raised by Reviewer 1. Although multidien periods are fairly stable over time within subjects (Supp. Fig. 5b, e), there is some variance or shift in period length, and this has the effect of attenuating the multidien peaks in periodograms averaged over long durations of recording. Still, the fact that small peaks rise above the background noise demonstrates that there is likely a rhythmicity (to some degree) in that frequency range. To illustrate this, Supp. Fig. 4 (previously Supp. Fig. 1) now shows that multidien peaks are more prominent in periodograms derived from shorter segments of recording (blue lines).

I don't understand how the slow frequency of interictal activity was calculated.

We apologize that our signal processing methods were not more clearly described. We have now included a new Supp. Fig. 2 that illustrates how wavelet decomposition was used to identify slow IEA frequencies.

From the graphs, it seems very variable within patients.

Multidien periods were variable across subjects (see new Fig. 2c, which shows that the distribution of multidien periods is broad compared to the circadian period) but relatively stable within subjects (Supp. Fig. 5b,e show that the autocorrelation for individual subjects remains high even over long periods of time). We have now added to Supp. Fig. 4 (previously Supp. Fig. 1) insets for several subjects showing detection count recordings separated by up to 7 years (e.g. Subject S10), highlighting the striking stability of multidien periods.

Why do you want to force an oscillation in this slow rhythm? Would it be more simple to look at interictal spike dynamics (acceleration?). I am not convinced that the authors could extract a rhythm in all patients (except fig 1). You need a much better detailed description and validation of what you did.

Looking at interictal spike rate acceleration would be simple but would likely be dominated by circadian spike oscillations (see raw detection count plot shown in Fig. 1d), and this strong variability would mask multidien rhythms. Wavelet analysis does not force an oscillation onto the data; rather, it decomposes the signal into component sinusoids and allows detection of slow rhythms by revealing power peaks in frequencies that would otherwise be difficult to observe. To clarify our methods, we now include a new Supp. Fig. 2 that illustrates wavelet decomposition. Panels (d)–(f) of this Figure validate the decomposition process by showing that the original data can be reconstructed by combining oscillations at different frequencies, and that the peak (innate) rhythms make up much of the original data.

The cluster analysis does not make sense to me. You should do it on an individual basis and not lump the results.

Individual periodograms are shown for all subjects (Supp. Fig. 4). Unsupervised clustering of these periodograms based on coefficients of principal components (PC's; Fig. 2b) was done primarily for visualization purposes to illustrate common features. Still, to provide more detail regarding potential variations, we include Supp. Fig. 7, which shows the first six PC's between all subjects. For example, PC#1 shows a peak at 28 days that accounts for 84.3% of the variance in average periodograms, indicating that oscillations of this period length are prevalent in our study population. In addition, we have now added histograms, stratified by gender (Fig. 2c), to illustrate the distribution of multidien periods across individual subjects.

Each patient appears unique based on fig 4. In fig 4, you should calculate the stat significance for each bin.

We agree that the shapes of seizure probability distributions differ across subjects (Fig. 4c,d). Our intent is simply to illustrate: (1) the phase distributions that lend to the circular statistic, and (2) the mean phase angle for seizure occurrence (vertical line). We are not comparing the normalized seizure counts at each phase angle bin across subjects, as these would undoubtedly differ across patients and have no bearing on the strength of individual seizure phase-locking to IEA rhythms. We have emphasized this in the Figure legend.

Also, you need to use surrogate analysis to demonstrate significance. You tested H0, but I would test H1 as well.

A surrogate analysis using shuffled data for an H0 distribution is technically feasible, but we would argue that this would add unnecessary complexity since the Omnibus test already tests the alternative hypothesis (a non-random tendency toward a one or more phase angles) against a null hypothesis, which is a uniform circular distribution. We have now clarified this in the Statistics section.

All considered, the title is misleading, you do not show phase locking, you show phase modulation, as the majority of seizures are outside the bins of the preferred phases in many patients. Phase locking has no strict definition in terms of how much of the events must be in a specific phase, but for place cells, the firing phase is nicely predictive of the spatial position of the animal.

We agree with this critique. Our intent was not to be misleading, as "phase locking" has no strict definition, but nevertheless we have revised the title to use the term "modulate" instead of "locked to".

4. Was neurostimulation always on and continuous? If not, you need to re-analyze the data to take into account this factor.

We thank the Reviewer for this comment, which echoes a concern raised by Reviewer 1. We agree that brain stimulation could potentially influence IEA and seizure patterns. As a result, we have now included an illustration of multidien rhythms during periods when the stimulation function of the device was turned OFF in a few subjects (Supp. Fig. 4, green lines). This shows that IEA rhythms continue during these periods, strongly arguing that stimulation is not directly responsible for these rhythms. Please see additional comments on this point enumerated in our response to Reviewer 1.

Of note, the RNS System is a closed-loop device, so stimulation is not continuous or even scheduled intermittent, but rather delivered on-demand in response to detection of abnormal patterns of activity.

In summary, I think that the paper could influence thinking in the field of epilepsy. However, the authors failed to convince me. I asked statisticians of my team, and they were puzzled by the unnecessary complexity of the approach (they were not convinced either). I would be delighted to be convinced. I recommend giving the authors another chance.

We agree that our findings could be influential in both research and clinical realms of the field of epilepsy. We also agree that unnecessary complexity should be avoided, and we sincerely hope that the additional analyses and clarifications included in the revised manuscript are more convincing to the Reviewer. We maintain that our approach reflects appropriate mathematical rigor for this type of data, and that it was essential for uncovering relationships between seizures and multidien rhythms in chronic ECoG datasets⁷.

Reviewer #3:

This ms from Baud and colleagues describes long-term recordings from epileptic patients which suggest that the frequency of intracranially recorded interictal events varies with a multi-day as well as an intra-day rhythmicity. Furthermore, in recorded patients, the occurrence of ictal events seems to be linked to a specific phase of the multi-day rhythms. These hypotheses are quite well supported by the data presented, even if the wavelet fit to time sequences is not so convincing. It is a shame that the study did not include a predictive element. The data should be of interest to the wider community and poses several interesting questions for future work. What is the basis for the slow rhythm and how does it influence seizure generation?

Specific points:

1. The slow rhythm shown in Fig. 1d does not really have a sinusoidal aspect. Superimposed on the diurnal changes, this data seems rather to show a relatively sudden (2-4 days) increase in the frequency of hourly counts which is followed by a much slower decay (10-15 days). So how exactly was the orange multidien curve derived? Was this pattern typical for all the patients? How could the conclusions change with a more realistic curve fitting?

Our study's methodology involves wavelet decomposition, an advanced but widely used signal processing technique that detects obvious and more subtle rhythmicities which would not be

characterized or quantified as effectively using simple curve-fitting approaches. Please also refer to responses to Reviewer 2, who raised similar concerns. We now include a new Supp. Fig. 2 that clearly illustrates the steps involved in wavelet decomposition. These methods are useful for estimating the spectral characteristics of a non-stationary signal in the time domain (our data being hourly IEA detection counts over months to years), and for analyzing how these underlying periodic components of a signal may also change over time²¹.

The same analysis was applied to all subjects, and individual results are shown in Supp. Fig. 4. All subjects demonstrated a robust circadian peak and most subjects also had a multidien peak. Multidien period length was variable across subjects (shown in new Fig. 2c) but relatively stable within subjects, even over long recording durations (see autocorrelation analysis in Supp. Fig. 5b,e; also see new insets in Supp. Fig. 4 showing, in several subjects, detection count recordings separated by up to 7 years (e.g. Subject S10) with striking stability of multidien periods). Still, there is some within-subject variance in period length, and this has the effect of attenuating the multidien peaks in periodograms averaged over long durations of recording. To illustrate this, Supp. Fig. 4 now shows that the prominent multidien peaks are more easily observed in periodograms derived from shorter segments of recording (blue lines).

Of note, the sudden increase and slow decrease in the multidien oscillation apparent in Fig. 1d is not a consistent feature across other subjects (for example, see Subject S6 in Supp. Fig. 4).

2. In the year-long recording of Fig. 1f, there seem to be some amplitude constraint on the orange curve, such that it misses the peaks of the highest daily counts (April-June). What is it? Was there a similar variability in amplitude of multi-day peaks in interictal frequency for other patients?

The orange curve is just one component of the raw signal that is derived from wavelet decomposition (the dominant slow oscillation), and other components would better approximate other features of the signal, like the sharp peaks. For clarity, these other components are not depicted in Fig. 1. They are shown now in a new Supp. Fig. 2 that illustrates signal processing steps.

3. The phase window for seizure occurrence (Fig. 1g) depends on the orange curve. How does interpretation change if the orange curve does not reflect the data faithfully. In Dec 2015 and July 2016 for instance (Fig. 1f) multi-peaked changes in count number are reduced to a single sinusoid.

We apologize for the lack of clarity in our methodology. The orange curve faithfully represents one component of the data based on wavelet transform using a Morlet mother wavelet, and only frequency components that had peaks in the periodogram were used in the seizure phase analyses. Our signal processing methodology is described more thoroughly in the new Supp. Fig. 2. Panel **d** of this Figure shows how instantaneous phase (dashed red line) can be derived from the wavelet transform, and this was used for phase analysis of seizure timing. Moreover, this panel illustrates that component peak frequencies (determined by wavelet power analysis)

can be used to reconstruct most of the original data, thus corroborating faithful signal decomposition. Please also see our similar clarifications for Reviewers 1 and 2 above.

4. In a subset of women, seizures occur with frequency linked to the menstrual cycle. The ms notes carefully that men and women both exhibited multi-day periodicity in interictal event occurrence. Still it would be interesting to know if the study picked up any of this group of women with epilepsy.

As stated in response to a similar point raised by Reviewer 1, nearly all of the 37 subjects (22 males, 15 females) showed multidien periodicities, and the distributions of periods were similar in male and female subjects, as shown now in new Fig. 2c. Of the subset of 14 subjects fulfilling criteria for seizure analysis, 9 (64%) were male and 5 (36%) were female. Of the 5 female subjects included in the seizure analysis, one had catamenial seizure cycling based on clinical reports, and she was found to have IEA periods at 13 and 26 days. In other words, 14 out of 15 females in the study had no clinical evidence of catamenial seizure variation. Thus, while we cannot exclude a contribution of catamenial cycling, it seems unlikely to explain fully our observations about seizure timing. We now include these points in the text, and we indicate in Supp. Fig. 4 which female subject (S29, labeled '(c)') was thought to have catamenial epilepsy on clinical grounds.

5. Fig. 2. The parameter on the vertical axis might be better described. Amplitude of what?

We thank the Reviewer for identifying this ambiguity. Amplitude on the vertical axis represents the square root of spectrogram power. The axis label, Figure Legend, and Methods have been revised accordingly.

6. The data of Fig. 4b seems to show a phase specificity for the association between seizure occurrence and slow variations in counts of interictal events. As in Fig. 1 though it seems to be based on a poor sinusoid-like wavelet fit to changes in frequency of interictal events. If seizure occurrence depends on some state parameter associated with increasing interictal frequency, why not plot seizure occurrence against day-to-day changes in interictal event counts? The author's hypothesis posits that seizures are closely associated with day-to-day increases in interictal frequency.

Decomposition of time-series data into component sinusoidals by wavelet transform and use of instantaneous phase in seizure timing analyses are discussed in responses to other comments above.

The Reviewer alludes to the unique implications of our findings for identifying periods of heightened seizure risk. However, simply using day-to-day changes in interictal detection counts to anticipate seizures would ignore the influence of circadian rhythms. In the revised manuscript, we present new analyses that characterize the combined influence of multidien and circadian rhythms on seizure timing (Fig. 5, Supp. Fig. 8). By calculating a risk ratio based on the circadian and multidien phases of all seizures, we show that rhythms on both timescales

contribute to determining seizure risk. Multidien rhythms have the greater contribution, but addition of circadian rhythms increased the seizure risk ratio in all subjects.

7. On the possibility of prediction: Maybe as a next step, it could be informative to define a long-term seizure rhythm from say 12 months data, and then test the accuracy of predicted dates of seizure occurrence against real seizure events over the next 6 months. If the window of peak seizure susceptibility is 2-4 days per month, as this data suggests, verification of the accuracy of predictions might open the way to time-limited protective measures or treatments. Some variant of this approach might be discussed.

We completely agree. The new Fig. 5 shows a seizure risk ratio map, averaged across all subjects used in the seizure timing analysis, in the circadian versus multidien phase space. White lines depict the hypothetical trajectory of a subject with an 8-day multidien period through this space, showing that seizure risk would vary over time as a function of both underlying rhythms. Ideally, data like this would be prospectively validated, and, while these efforts are underway, they are beyond the scope of this initial descriptive study. We have now explicitly stated this goal for future work in the Discussion.

Reviewer #4:

This paper is a robust analysis of data from the RNS over many months in several patients, and shows how detected epileptiform events have periodicity on multiple time scales, and seizures are more likely during these periods. This is a very unique dataset that provides information about the long term seizure risk. The authors show not only that most patients (in this small dataset) have diurnal and multidien variation in the IEA, but also that seizures were more likely to occur at specific intervals in these variations. This information is of interest to the larger community, helps clinicians and scientists understand the long term variability of seizures, and helps answer a longstanding debate about the utility of interictal discharges. The statistical analysis is advanced, and an appropriate method to deal with these cyclical statistics (although they are likely unfamiliar to most readers). There are some concerns with the paper.

Storage limitations *One of the primary limitations of this RNS device is its very limited storage, and the method of saving data. The device only saves events that are designated for detection, and it only can save a limited number of those before starting to overflow and dropping earlier detections. The patients have to upload data regularly to assure detections are not lost. Thus this is a very limited dataset to begin with (only saving clips of detected events), with serious concerns about lost data (from when there is overflow or longer periods without an upload). These limitations are addressed to some degree in the paper, but they are not really explained. Thus this is not “10 years” of data, but 10 years in which a very limited subset of detected events may or may not have been uploaded.*

We thank the Reviewer for a thorough appraisal of our manuscript and the methodology therein. The Reviewer correctly points out that the storage capacity of the RNS System is limited and only 6 minutes of 4-channel raw ECoG (typically four 90-second ECoG tracings) can

be stored on the device at a given time. Raw ECoG tracings are obviously important clinically and they were used here for some of our analyses (see Methods section on PPV criteria). However, a critical point is that the bulk of the data analyzed here was not raw ECoG, but rather hourly detection counts and the dates/times of 'Long Episodes' (termed LEA in the manuscript), which are stored by the device for up to 28 days without being overwritten or otherwise lost. Thus, as long as subjects downloaded device data at least once every 28 days, the hourly detection counts and LEA are continuous with no loss of data. If nevertheless there were delays in downloading that resulted in gaps, we used a conservative gap interpolation approach (tailored by period duration), described in Supp. Fig. 2 and 3, to ensure continuity while avoiding misrepresentation of the data. As stated in the text, data with gaps longer than six days were considered discontinuous and analyzed in separate segments. We have added clarification on these points to the Methods text and elsewhere.

This leads to a couple of questions and a concern. How do the authors know there wasn't lost data from overflow? How can they be certain there weren't gaps in uploading (i.e. did patients really upload data daily for 10 years? That seems remarkable.) How can they be certain the uploading contained all data since the last upload? The paper should make it very clear that these limitations exist. (More questions on gaps below).

The Reviewer is correct that patients implanted for long periods of time eventually stop downloading device data on a daily basis as they are asked to do initially. However, as explained in the previous response, as long as they downloaded at least once every 28 days, the data used in our analysis was continuous. Although the vast majority of our data was continuous, some subjects did have gaps, and our conservative approach for handling gaps in data is described in the Methods.

The RNS only records what you tell it to record. This is important, because the only other device with similar data (the NeuroVista device) recorded ALL of the data, and thus this dataset is at a clear disadvantage. This limitation needs to be very clearly stated.

The NeuroVista device recorded ECoG continuously by virtue of telemetric transfer to an external unit with fewer storage constraints than the implanted RNS System. Thus, the data provided by the two devices are quite different. However, several points are worth noting:

- (1) As described above, we did not rely on stored ECoGs for identification of seizures; rather, we used mostly continuous hourly detection counts and dates/times of LEA in subjects for whom LEA was validated as a highly reliable proxy for seizures (see new Supp. Fig. 1 and detailed response to Reviewer 1 above),
- (2) The NeuroVista data was 6 months to 2 years in duration⁷, considerably shorter than the recording durations in our study (7 subjects with >100 months recording; Supp. Table 1),
- (3) In patients with bitemporal seizures, NeuroVista leads were implanted unilaterally "over the hemisphere that generated the most frequent, stereotypical seizures"⁶, which undoubtedly resulted in undetected contralateral seizures; by contrast, 17/37 of our subjects were implanted with bilateral leads, so seizures were likely more completely sampled in our subjects.

Still, because we report a novel use of RNS System data, we agree that it is essential to explicitly describe the limitations of the device. We have now added additional text that clearly states the limitations of the RNS System.

Algorithm limitations *That brings up another concern with what actually gets recorded. What is the sensitivity/specificity of these detections? (in these patients, I suspect it will not be possible to test this).*

Detection parameters were tuned by clinicians to optimally detect epileptiform activity. Since the device does not store continuous ECoG, it is not feasible to accurately determine the sensitivity and specificity of these detections. Importantly, though, analyses were performed separately on recording blocks with constant detection settings. We did test the specificity of LEA for electrographic seizures by visually inspecting and scoring ECoGs and calculating the positive predictive value for LEA being an electrographic seizure. Only subjects with high PPV (>90%) were included in the seizure timing analysis. This is now emphasized in the text and the methods are clarified in the new Supp. Fig. 1.

There is one surrogate for specificity--how often were there magnet swipes/reported symptoms that were NOT detected by the algorithm? How were these "false negatives" dealt with?

Analysis of ECoGs stored in response to patient magnet swipe is an interesting idea, and one that we considered initially. However, magnet-triggered ECoGs are severely limited for several reasons:

- (1) Magnet swipe generally allows storage of 60 seconds of ECoG prior to the swipe; thus, patients who had electroclinical seizures but swiped the magnet too late (say, 1.5 min after the seizure), would show no seizure activity on the stored ECoG (false negative).
- (2) Clinicians often advise patients to swipe the RNS magnet for the purposes of spell characterization (for instance, to determine whether intermittent headaches are related to seizure activity), even though the suspicion that these symptoms are seizures is very low. Such ECoGs typically show no seizure activity (true negative).
- (3) Some patients reliably swipe the magnet soon after onset of seizure symptoms, and review of the corresponding ECoGs shows clear seizure activity (true positive)
- (4) Patients with highly active interictal backgrounds can have epileptiform activity apparent on virtually all stored ECoGs, and, in magnet-triggered ECoGs, it can be difficult to discern the symptomatic patterns of activity from the asymptomatic interictal background activity (false positive)
- (5) Unlike detection counts and LEA, magnet-triggered ECoGs can be overwritten based on storage constraints (lost data)

Taken together, we felt that magnet-triggered ECoGs would not be sufficiently reliable for our analysis.

My guess is that algorithms were dynamically adjusted to improve sensitivity. But then, if the algorithms were changed to detect different events, how do you reconcile all the previously-undetected events that were not detected? The tone of this article implies an unchanging, infallible detector in each patient, which is really what is needed to make this study ideal. If the algorithms changed, this will need to be clearly stated and reconciled. And the paper should make it very clear what the limitations of the algorithm are.

In clinical practice, RNS detection algorithms are iteratively adjusted based on inspection of stored ECoGs to optimally detect desired patterns of epileptiform activity. Detection count analysis over long periods of time must account for these intermittent changes, and we apologize if the manuscript conveyed the sense of an ‘unchanging, infallible detector’. To the contrary, we describe in the Methods that blocks of data with stable detection settings were analyzed separately and normalized (z-score) before concatenation with other blocks. We also discarded the first few months of data because this is a period of time when detectors are typically changed rapidly and not at steady-state. For clarity, we now include a new Supp. Fig. 2 that depicts the process of normalization within recording blocks that have stable detection settings. We have expanded portions of the text that describe limitations of our approach to ensure these issues are understood by the readership.

The algorithm and RNS system also have their own unique nomenclature, which are not well explained. An “IEA” in the RNS can mean several things, but often is detecting abnormal discharges or runs of activity, rather than individual spike and waves. Most readers, however, are going to assume “IEA” refers to basic spikes—is that so? Are the authors claiming that they have detected all spikes? If not this needs to be clearly stated (the paper really does imply that that is what is happening, especially when bringing up the controversy of spike rate and seizure risk). There needs to be a clear section defining how IEA are chosen, what typical IEA entailed, and what types of IEA might not be detectable with RNS, how many spikewaves are/are not detected, etc. Figure 1 seems to imply it is detecting spikewaves, but in this reviewer’s experience with RNS that is not what is typically detected.

The Reviewer correctly notes that IEA detected by the RNS System is not always spike-wave discharges, and we do not claim to detect all spikes (see discussion of detection sensitivity/specificity in a previous response). However, IEA typically falls into one of three categories: spiking, rhythmic alpha/beta, and low-voltage fast activity. These patterns of IEA commonly detected by the RNS System have been described previously³, but, for clarity, we have added a new Supp. Fig. 1 that provides examples of the types of IEA detected by the RNS System. The Figure also explains that the rationale for calculating a PPV for LEA being seizures relates to the fact that epochs of abundant but poorly organized interictal spiking are sometimes recorded by the device as LEA but are not electrographic seizures.

Use of the word “random” *One of the main results of the paper is that seizures are more or less likely at different times: Intro: “Our findings indicate that seizures are not random events and that slow rhythms of IEA are a critical biomarker for seizure prediction.” Also p. 6 “seizures are non-random events organized by underlying biological rhythms operating over long timescales.”*

These statements are not exactly true and should be said with more care. Seizures are in fact still ‘random’—on any given time scale they are not deterministic, and there is no time when they can be predicted perfectly. What is happening is that there are times when they are more likely, and it would be easier to predict. What the authors are showing is that the underlying RISK is variable, with individual seizures still occurring at random times (albeit affected by a variable risk), very similar to past modeling work shown in Jirsa et al Brain 2014 and several articles by Lopes da Silva.

We thank the Reviewer for these insightful points with which we agree completely. We have rephrased the relevant portions of the manuscript to clarify that our results do not resolve the stochastic nature of seizures. We include additional analyses (Fig. 5, Supp. Fig. 8) showing that circadian and multidien rhythms are factors that contribute to determining seizure risk. We revised the manuscript title to include the term ‘seizure risk’ so as not to be misleading about potential for seizure prediction. Finally, we now include references to work by Jirsa and Lopes da Silva.

Also the phrase “biomarker for seizure prediction” is too ambiguous. It is really a biomarker of seizure risk, which then can be used to inform seizure prediction.

Agreed. We have modified the text accordingly.

References Past work In addition to the modeling work above, there have been other physiological studies showing how EEG can change over longer time periods to suggest ictogenicity. For instance, Huberfeld et al. Nat Neurosci 2011, Karafin et al Seizure 2010, Chauviere et al Ann Neurol 2012).

These references^{8,22,23}, which help contextualize our findings, have been added to the manuscript. We are grateful for the suggestion to include them.

Proving hypotheses. The statement that “For example, seizures preferentially occur during the rising phase of a multidien IEA cycle, but this could coincide with the peak or the trough of the circadian IEA cycle, as observed by others, and shorter timescale studies would draw seemingly contradictory conclusions” is very intriguing. However, could the authors please show if and when that happens? I don’t know of another time when similar data will be presented—this paper has the data ready right now! Please inform us about whether the ultradian and circadian phases actually have such a phase mismatch, and how often it occurs. If it does not happen in your data, then this statement is probably incorrect.

To directly address this question, we now include a new Supp. Fig. 8 with scatterplots of circadian versus multidien phase at time of seizures. In Subject S33, for example, seizures align strongly with a given multidien phase (just before peak) but occur over a wide range of circadian phases. Of note, however, some subjects demonstrate relatively stronger modulation of seizure timing by circadian phase (e.g. subject S37). Most subjects demonstrate influence of both underlying rhythms (e.g. subject S24). By calculating a risk ratio based on the circadian and

multidien phases of all seizures, we are able to show that rhythms on both timescales contribute to determining seizure risk (Supp. Fig. 8, Fig. 5).

Prediction

In addition, the authors talk about how this would help with prediction. This is never proven and there is never any attempt to do so. It is probably outside the realm of this paper, but care should be taken with such statements—they have not proven that it will have any bearing on seizure prediction.

We agree that the data presented here do not directly support a role for multidien rhythms in seizure prediction. We mention seizure prediction only to highlight the potential utility of the long timescale rhythms we have characterized. As described above, we have now taken care to avoid conveying the impression that we have eliminated the random nature of seizures.

The new Fig. 5 shows a seizure risk ratio map, averaged across all subjects used in the seizure timing analysis, in the circadian versus multidien phase space. White lines depict the hypothetical trajectory of a subject with an 8-day multidien period through this space, showing that seizure risk would vary over time as a function of both underlying rhythms. Ideally, data like this would be prospectively validated, and, while these efforts are underway, they are beyond the scope of this study. We expressly state that prospective seizure prediction remains a critical goal for the future (“leveraging knowledge of subject-specific circadian and multidien rhythms for prospective seizure prediction remains a major goal of future work”).

Gap filling is very important with this device, and is a clever way to deal with potential problems in the data. But some things are still not clear. How can we be sure whether events were discarded? What type of spline was used for interpolation; was it always monotonic?

The spline was always monotonic, and interpolation methodology is described in detail in the Methods and Supp. Fig. 3. For the majority of subjects, gaps in the recordings accounted for a small percentage of overall data (Supp. Table 1) and are therefore unlikely to significantly influence the main results.

What was the “cone of influence”? That was not clear.

The ‘cone of influence’ (COI) is the region of the wavelet spectrum, shaped according to period length, in which edge effects impede accurate periodic estimation (therefore these sections are not included for analysis). COI is now more completely described and graphically depicted in a new Supp. Fig. 2 illustrating signal processing steps.

Furthermore, looking at the data it is not clear how these tools are being used. For instance in sup Table1, it says patient S13 has 73% valid data, but looking at Supp Fig 1 it is not clear where the gaps are (and in that figure I cannot make out where any of the “interpolated” purple segments are).

Subject S13 had 116 months of total data but, for clarity, that Supplementary Figure (now Supp. Fig. 4) only shows two years' worth of data, which is why all gaps are not visible. In addition to additional text and illustrations regarding methodology, the new Supp. Fig. 2 shows an example of a gap with a more direct illustration of how certain frequencies were handled using our gap criteria, including COI and interpolation.

Discernibility of differences. Fig 4 shows the actual differences in phase, but I am not sure all of those patients had a clear difference. Why is the amplitude of the difference smaller with larger N in the multidien (S4, 26, 7, 9 are all the highest N, and all have the “flattest” distributions)? That is concerning—the difference appears trivial in all the patients with the highest N (which is usually 1-2 orders of magnitude higher than the others). With such high N, “statistical significance” is somewhat guaranteed, but is there actually a discernable difference? I doubt that multidien variations would be helpful in those patients, as there just isn't enough of a difference to measure. And if these distributions tend to “flatten” with longer recordings, will discernability vanish for all patients? This is actually probably the most important concern in the paper.

By definition, subjects with the highest N have the highest seizure counts. For these subjects, we agree that the circular histograms of seizure frequency are more evenly distributed over multiple phase angles than for subjects with lower seizure burden (Fig. 4c,d). To address the Reviewer's concern, we offer several observations and additional data:

(1) While subject S4 does have a relatively flat distribution, subjects S26, S7, and S9 have clear peaks which, if anything, are more compelling given the high N. In other words, if seizures were not related to underlying multidien rhythms, we might not expect to see even as much of a peak as we do in subject S9 with $N > 100,000$.

(2) We agree that multidien variations might be less helpful in subjects with very frequent seizures. To take the extreme: In a subject who has one seizure every hour, using multidien rhythms to identify days of increased seizure risk would be meaningless; by contrast, subjects with low or intermediate seizure frequencies could potentially benefit from using circadian and/or multidien rhythms to identify periods of heightened seizure risk.

(3) Consistent with this, we showed that the strength of multidien locking across subjects is inversely proportional to seizure frequency (previously Supp. Fig. 5c, now Supp. Fig. 9c).

We performed additional analyses to obtain a more intuitive estimate of the effect size based on risk ratios. We discretized the phase variable in two categories: in-phase or anti-phase with the preferred circadian or multidien angle for each individual patient. Considering these categories as the result of a ‘test’, we defined true positives as a seizure occurring while in-phase with the underlying circadian AND/OR multidien rhythm. A false negative was defined as a seizure occurring anti-phase with the underlying circadian AND multidien rhythm. False positives were all the hourly circadian OR multidien in-phases when seizures did not occur. True negatives were all the hourly circadian AND multidien anti-phases when seizures did not occur.

This allowed calculation of a risk ratio for seizures ($\frac{TP/(TP+FP)}{FN/(TN+FN)}$). When combining phase information from the underlying circadian and multidien rhythms, we found small (RR 1.2, 95% CI: 1.1-1.3) to very large (RR 24.5, 95% CI: 3.4-175.9) effect sizes in subjects with high and low seizure rates, respectively, and a large effect size summary across subjects (RR 6.0, 95% CI: 3.6-10.2, Supplementary Fig. 8).

Also, there is no legend for the X axis in that figure.

The x-axis for Fig. 4 panels (previously (c)–(f), now (c) and (d)) are phase angle. This has been added to the Figure for clarity.

REFERENCES

1. Spencer, S.S., Goncharova, I., Duckrow, R.B., Novotny, E.J. & Zaveri, H.P. Interictal spikes on intracranial recording: behavior, physiology, and implications. *Epilepsia* **49**, 1881-1892 (2008).
2. Janszky, J., *et al.* Spatiotemporal relationship between seizure activity and interictal spikes in temporal lobe epilepsy. *Epilepsy Res* **47**, 179-188 (2001).
3. Spencer, D.C., *et al.* Circadian and ultradian patterns of epileptiform discharges differ by seizure-onset location during long-term ambulatory intracranial monitoring. *Epilepsia* **57**, 1495-1502 (2016).
4. Sun, F.T. & Morrell, M.J. The RNS System: responsive cortical stimulation for the treatment of refractory partial epilepsy. *Expert Rev Med Devices* **11**, 563-572 (2014).
5. Morrell, M.J. & Halpern, C. Responsive Direct Brain Stimulation for Epilepsy. *Neurosurg Clin N Am* **27**, 111-121 (2016).
6. Cook, M.J., *et al.* Prediction of seizure likelihood with a long-term, implanted seizure advisory system in patients with drug-resistant epilepsy: a first-in-man study. *Lancet Neurol* **12**, 563-571 (2013).
7. Karoly, P.J., *et al.* Interictal spikes and epileptic seizures: their relationship and underlying rhythmicity. *Brain* **139**, 1066-1078 (2016).
8. Chauviere, L., *et al.* Changes in interictal spike features precede the onset of temporal lobe epilepsy. *Ann Neurol* **71**, 805-814 (2012).
9. Krishnan, B., *et al.* A novel spatiotemporal analysis of peri-ictal spiking to probe the relation of spikes and seizures in epilepsy. *Ann Biomed Eng* **42**, 1606-1617 (2014).
10. Hoppe, C., Poepel, A. & Elger, C.E. Epilepsy: accuracy of patient seizure counts. *Arch Neurol* **64**, 1595-1599 (2007).
11. Sun, F.T. & Morrell, M.J. Closed-loop neurostimulation: the clinical experience. *Neurotherapeutics* **11**, 553-563 (2014).
12. Fontolan, L., Morillon, B., Liegeois-Chauvel, C. & Giraud, A.L. The contribution of frequency-specific activity to hierarchical information processing in the human auditory cortex. *Nat Commun* **5**, 4694 (2014).

13. Zhang, Z., Telesford, Q.K., Giusti, C., Lim, K.O. & Bassett, D.S. Choosing Wavelet Methods, Filters, and Lengths for Functional Brain Network Construction. *PLoS One* **11**, e0157243 (2016).
14. Gadhomi, K., Lina, J.M. & Gotman, J. Discriminating preictal and interictal states in patients with temporal lobe epilepsy using wavelet analysis of intracerebral EEG. *Clin Neurophysiol* **123**, 1906-1916 (2012).
15. Richard, C.D., *et al.* SWDreader: a wavelet-based algorithm using spectral phase to characterize spike-wave morphological variation in genetic models of absence epilepsy. *J Neurosci Methods* **242**, 127-140 (2015).
16. Faust, O., Acharya, U.R., Adeli, H. & Adeli, A. Wavelet-based EEG processing for computer-aided seizure detection and epilepsy diagnosis. *Seizure* **26**, 56-64 (2015).
17. Ortiz-Rosario, A., Adeli, H. & Buford, J.A. Wavelet methodology to improve single unit isolation in primary motor cortex cells. *J Neurosci Methods* **246**, 106-118 (2015).
18. Moberg, A., *et al.* Highly variable Northern Hemisphere temperatures reconstructed from low- and high-resolution proxy data. *Nature* **433**, 613-617 (2005).
19. Cazelles, B., Chavez, M., Magny, G.C., Guegan, J.F. & Hales, S. Time-dependent spectral analysis of epidemiological time-series with wavelets. *J R Soc Interface* **4**, 625-636 (2007).
20. Ramsey, J.B. Wavelets in Economics and Finance: Past and Future. *Studies in Nonlinear Dynamics & Econometrics* **6**(2002).
21. Cazelles, B., *et al.* Wavelet analysis of ecological time series. *Oecologia* **156**, 287-304 (2008).
22. Huberfeld, G., *et al.* Glutamatergic pre-ictal discharges emerge at the transition to seizure in human epilepsy. *Nat Neurosci* **14**, 627-634 (2011).
23. Karafin, M., St Louis, E.K., Zimmerman, M.B., Sparks, J.D. & Granner, M.A. Bimodal ultradian seizure periodicity in human mesial temporal lobe epilepsy. *Seizure* **19**, 347-351 (2010).

Reviewers' comments:

Reviewer #1 (Remarks to the Author):

1. My pdf was missing supplemental figures 6-9
2. Need to remove phrase "tightly coupled" from abstract – an RR of 1.2 in patients with most seizures does not indicate tight coupling
3. Need to include the information regarding relative risk (average and range, as described in the penultimate author response to R4) in the abstract

Reviewer #2 (Remarks to the Author):

The authors did a good job answering many questions, and in clarifying methods.

There are a few points that remain to be clarified.

1. Maybe you can strengthen the interpretation by looking more carefully at the data. In Fig 1f, around July, there are two slow cycles, but each is composed of two faster IEA cycles. If you take them into consideration seizures occur during the rising phase (in line with your hypothesis). But smoothing them as you did (orange curve) produces a more widespread distribution over the cycle. How general is this. You may obtain better phase entrainment if you consider faster cycles?
2. I still have this issue with phase locking, which has a strict meaning in engineering, optics etc. We can accept a small jitter, but the data shows a very large dispersion. You do demonstrate a phase preference, and phase entrainment, but not real phase-locking. Either you remove phase locking from the ms, or develop more in the discussion that there is a wide dispersion (but see point 1 which may provide less dispersion).
3. Your Fig 4e suggests that phase preference decreases with the number of seizures. Is this a general phenomenon? This is clinically important, since in conditions of low seizure frequency, we may rely on the rising phase of the slow oscillation to give an alert signal.
4. Along these lines, the authors did not answer one of my questions also raised by another reviewer. Is there a predictive value? You can perform the analysis on several months, and use the rest of the dataset to see whether a warning signal would be useful. I guess not given the dispersion, but this is a key information that needs to be provided. It is OK if there is no predictive value, it is interesting to know about the multidienn. The risk factor analysis is not sufficient. The latter is also misleading as patients with a lot of seizures (supplemental Fig 8) have a high risk everywhere. You cannot average individuals, calculating the risk factor is OK for individuals.
5. From Supp Fig 4, it is clear that rhythms vary a lot in time. You should quantify it in individuals. The autocorrelogram is widespread in the slow frequency range. The analysis now shown on shorter time scales better shows the peaks.

With a last effort, we can have an excellent story here.

Reviewer #3 (Remarks to the Author):

This resubmitted ms from Baud and colleagues suggests there is a multi-day and an intra-day rhythmicity, in the timing of interictal events recorded from ambulatory patients with epilepsy. Furthermore, ictal events seem to occur on a specific phase of the multi-day rhythms. These are useful data. My questions mostly concerned the fitting of the data. Responses clarify some issues, but sometimes avoid the question.

Specific points:

1. I was concerned about the goodness of fit of the wavelet based analysis of periodicity in inter-ictal frequency. In response the authors show in Supplementary Fig. 2 more details on how the

wavelet analysis was done and in Supplementary Fig. 4 most, possibly all, of their data. This will help readers.

The question on the shape of the oscillations remains. The example in Fig. 1d seems to show a sudden increase followed by a slower decline. Some but not all of the data in Supplementary Fig. 4 seems to show comparable behavior. Did the authors attempt to derive a goodness-of-fit parameter for this approach? Presumably the more wavelet components the better the fit. How did they decide when to stop?

Supplementary Fig. shows analyses of subselected data (3-6 months) for several of the data-sets. Often components at longer periodicities are enhanced. To sustain the hypothesis of a specific long-term rhythmicity it seem important to show that several subselected periods from the same patient give similar long-terms peaks of periodicity.

2. The orange curve in Fig. 1f remains a poor estimate of the daily counts trace since higher frequencies are missing. Supplementary Fig. 2 helps, but does not seem to correspond to the period shown in Fig. 1f, which might be improved by showing another critical higher frequency component.

3. The reply seems to miss the question. Was a single (orange) waveform used as the basis to extract a phase relation of seizure to interictal frequency? How does the phase relation change if a better, possibly multiple-wavelet fit was used?

6. This reply also misses the question. Fig. 4b suggest that ictal events occur when interictal frequency is increasing, whether the period is 10, 20 or 30 days. If so, there is a simple way to check the data and also to support the idea that some seizure-related state parameter depend on interictal frequency. Seizures should occur only when the day-to-day interictal frequency is increasing. This approach would not ignore the circadian fluctuations, since day-to-day frequency may increase over several consecutive days. Presumably ictal events would be very probable.

Reviewer #4 (Remarks to the Author):

The authors have done extensive modifications and been very responsive to the criticisms. This paper now is much more convincing, and is a result that I am very interested in. It has novel analysis and unprecedented data. I have just a few comments that need clarification.

IEA need to be defined earlier and explicitly. There should be something in Results right after "Subjects" that defines it briefly. And it should again be described explicitly (i.e. "for this device, IEA is defined by...."). The problem is that IEA in Neuropace is essentially a proprietary definition. And additionally it is unique for every patient, and can be changed by the user. This is a bigger deal than the authors make it—it is likely that some of the other reviewers of this paper actually misunderstood this as well. The reader is most likely going to believe it means either a traditional spike wave discharge and/or a clear electrographic seizure. And the reader is going to believe that you are claiming to have detected all of them. Thus it is imperative to give an objective definition, followed by an explanation of the data that are saved.

The other problem is that the authors themselves are not consistent when referring to IEA. The definition of IEA versus "IEA rates used in the analysis" are ambiguous and used interchangeably. The abstract states that seizures are "preferentially occurring near the peak of IEA." I think you mean peak of IEA rates? (the "peak of the IEA" would be the instant the spike is maximum amplitude). There is also a section in results called "Phase analysis of IEA peak..." It is not the IEA peak, but the peak of the IEA rates. Another section reads "underlying IEA rhythms" Again, you are not analyzing the IEA rhythm, but the periodicity of the peak rates. This style needs to be policed throughout the whole paper.

In Supp Fig 1a, a statement should be added reminding the reader that these data are not used in the analysis, just the counts of these data.

I suggest a simple change to line 63: "wavelet transform to resolve component rhythmicity of the IEA rate". This is because many readers will assume you are decomposing the actual IEA waveforms, since that is what is almost invariably done with wavelets and EEG. I also suggest a

stronger “sell” of Suppl Fig 2—this is really a very nice figure to help understand what is going on. Supp Fig 8 and Fig 5 are beautiful. I’ve never seen data like this before—this should be very impactful, as it shows some patients that clearly have variability of seizure risk on two different time scales. Remarkable! But I do not think the authors explained these results well enough in the text. It is somewhat buried in a jargon-rich section titled “Phase-space analysis of seizure risk”. I strongly suggest making this more of a highlight of the paper, accentuating it with more user-friendly title and explanation, and probably adding it to the abstract. A section title along the lines of “seizure risk variability on circadian and multidienn time scales” (or anything like that) would be helpful, as would a written presentation that is more practical and less buried in details. I suspect this result will be the one that is most often cited in this paper by the general audience (who are not going to be familiar with “phase-space”). Making it easier to find and understand is important.

The new analysis on risk ratios is very helpful to see which patients have discernable changes.

Typo line 101: “across in these subjects”

Reviewer # 1:

1. My pdf was missing supplemental figures 6-9

We regret that the Reviewer was not able to view these Supplementary Figures. We hope that the Editorial team can assist and that this Reviewer finds these data as informative as other Reviewers have found them.

2. Need to remove phrase “tightly coupled” from abstract – an RR of 1.2 in patients with most seizures does not indicate tight coupling

We have removed the phrase “tightly coupled” from the Abstract and Introduction.

3. Need to include the information regarding relative risk (average and range, as described in the penultimate author response to R4) in the abstract

We now include this information regarding relative risk in the Abstract.

Reviewer # 2:

The authors did a good job answering many questions, and in clarifying methods.

There are a few points that remain to be clarified.

1. Maybe you can strengthen the interpretation by looking more carefully at the data. In Fig 1f, around July, there are two slow cycles, but each is composed of two faster IEA cycles. If you take them into consideration seizures occur during the rising phase (in line with you hypothesis). But smoothing them as you did (orange curve) produces a more widespread distribution over the cycle. How general is this. You may obtain better phase entrainment if you consider faster cycles?

We agree with the Reviewer that faster IEA cycles are present in the data, as depicted in Supplementary Figs. 2 and 4. In Fig. 1, we previously showed only one exemplar slow oscillation to highlight a central finding of our work. However, we have now revised Fig. 1 to show this subject’s two main multidien (10-day and 26-day) IEA cycles, which together capture more features of the raw signal. Both of these multidien rhythms show phase preference for seizures, consistent with our central discovery, and variance in the phase domain is similar. Determining the combination of periodicities that optimally explains seizure timing (e.g. via periodic constructive interference) is beyond the scope of this initial study but remains a major goal of future work. Nevertheless, Fig. 5 and Supplementary Fig. 8 mark the beginning of these investigations, combining circadian and multidien frequencies to improve phase entrainment.

2. I still have this issue with phase locking, which has a strict meaning in engineering, optics etc. We can accept a small jitter, but the data shows a very large dispersion. You do demonstrate a phase preference, and phase entrainment, but not real phase-locking. Either you remove phase locking from the ms, or develop more in the discussion that there is a wide dispersion (but see point 1 which may provide less dispersion).

The term 'locking' has been removed from the manuscript text (except in instances of the statistical term, 'Phase Locking Value (PLV),' equivalent to the resultant vector length), from Figs. 4 and 5, and from Supplementary Figs. 8 and 9.

3. Your Fig 4e suggests that phase preference decreases with the number of seizures. Is this a general phenomenon? This is clinically important, since in conditions of low seizure frequency, we may rely on the rising phase of the slow oscillation to give an alert signal.

Fig. 4e shows that the preferred phase (vector angle) is relatively stable over long periods of time but that the strength of phase entrainment (vector length) actually increases as seizure frequency decreases in this subject. For clarity, we have now re-worded a sentence in the Results corresponding to Supplementary Fig. 9c where we quantify this relationship across subjects. In that Figure, we show that the strength of phase entrainment to multidien rhythms is inversely proportional to seizure frequency. Thus, it does seem to be a general phenomenon that in subjects with a high number of seizures, seizures are more 'decoupled' from the preferred phase than in subjects with a lower number of seizures. We agree with the Reviewer that this observation may have clinical implications, and we have edited a sentence in the Discussion to reflect this: "Multidien and circadian rhythms may be most predictive in subjects with a low or moderate seizure rate where phase preference is highest."

4. Along these lines, the authors did not answer one of my questions also raised by another reviewer. Is there a predictive value? You can perform the analysis on several months, and use the rest of the dataset to see whether a warning signal would be useful. I guess not given the dispersion, but this is a key information that needs to be provided. It is OK if there is no predictive value, it is interesting to know about the multidien.

The aim of our study was to use a unique dataset to characterize IEA rhythms on multiple timescales and to investigate their relationships to seizure timing. The predictive value of our findings is obviously critical for clinical applications, but we feel this is best addressed in a prospective manner rather than training individualized classifiers on retrospective data. Prospective seizure prediction remains a major goal for future work, as stated above and in the Discussion, but this may require development of novel statistical approaches (see responses to Reviewer 3 below). Indeed, other Reviewers have acknowledged that seizure prediction is a substantial challenge that is more appropriate for a follow-up prospective study (Reviewer 4: "[Seizure prediction] is probably outside the realm of this paper"; Reviewer 3: "On the possibility of prediction: Maybe as a next step, it could be informative to define a long-term seizure rhythm

from say 12 months data, and then test the accuracy of predicted dates of seizure occurrence against real seizure events over the next 6 months.”).

The risk factor analysis is not sufficient. The latter is also misleading as patients with a lot of seizures (supplemental Fig 8) have a high risk everywhere.

Subjects with a very high frequency of seizures do indeed have a high risk of seizures at all times, and this is reflected in the risk ratio plots (Supplementary Fig. 8) as well as in the decreasing PLV with increasing seizure rate (Supplementary Fig. 9c). As stated in the Discussion, multidienn and circadian rhythms may be most predictive in subjects with a low or moderate seizure rate.

You cannot average individuals, calculating the risk factor is OK for individuals.

We agree that risk should be assessed at the individual level for prospective seizure prediction, and we provide individual risk ratios in Supplementary Fig. 8. The purpose of averaging risk ratio maps across subjects (Fig. 5), after alignment to the preferred phase, is to provide an effect summary and to highlight the remarkable consistency of seizure timing in relation to underlying circadian and multidienn phase, despite the variability in circadian timing and multidienn period lengths across subjects (Fig. 2). To take into account the fact that a different number of observations contributed to each individual map, we weighted the average, as is commonly done in meta-analyses.

5. From Supp Fig 4, it is clear that rhythms vary a lot in time. You should quantify it in individuals. The autocorrelogram is widespread in the slow frequency range. The analysis now shown on shorter time scales better shows the peaks.

Supplementary Fig. 4 indeed shows some within-subject variance in period length that has the effect of attenuating multidienn peaks in periodograms averaged over long durations of recording. As pointed out by the Reviewer, periodograms derived from shorter segments of recording (blue lines in Supplementary Fig. 4) show more prominent multidienn peaks. The autocorrelation for individual subjects is quantified in Supplementary Fig. 5e. In addition, we now quantify the variance in multidienn periodicities in individuals and report this as an average wavelength \pm standard deviation in Supplementary Fig. 4. We have revised the Methods accordingly. Despite some variance, we feel that the observed degree of stability is impressive given the long recording durations and the fact that subjects were not studied under controlled conditions but rather ‘real world’ clinical circumstances.

With a last effort, we can have an excellent story here.

Reviewer # 3:

This resubmitted ms from Baud and colleagues suggests there is a multi-day and an intra-day

rhythmicity, in the timing of interictal events recorded from ambulatory patients with epilepsy. Furthermore, ictal events seem to occur on a specific phase of the multi-day rhythms. These are useful data. My questions mostly concerned the fitting of the data. Responses clarify some issues, but sometimes avoid the question.

Specific points:

1. I was concerned about the goodness of fit of the wavelet based analysis of periodicity in inter-ictal frequency. In response the authors show in Supplementary Fig. 2 more details on how the wavelet analysis was done and in Supplementary Fig. 4 most, possibly all, of their data. This will help readers.

The question on the shape of the oscillations remains. The example in Fig. 1d seems to show a sudden increase followed by a slower decline. Some but not all of the data in Supplementary Fig. 4 seems to show comparable behavior. Did the authors attempt to derive a goodness-of-fit parameter for this approach?

The Reviewer makes an observation regarding the morphology of the periodic multidien fluctuations in Fig. 1, and we agree that this pattern is not present in all subjects (see, for example, subject S6 in Supplementary Fig. 4). The rise and decay slopes result from combination of rhythms at different periods (scales) taken into account by wavelet decomposition. To estimate goodness-of-fit, we have now calculated the Pearson correlation coefficient for wavelet-based approximation of IEA time series for all frequencies from 2d-45d (Fig. 1f, gray curve). The result ($r = 0.93$) reveals excellent correlation between the original and reconstructed time series.

Presumably the more wavelet components the better the fit.

As can be seen in the new version of Fig. 1, adding wavelet components does indeed improve approximation of the original data.

How did they decide when to stop?

We used 89 period bins (scales) spanning 2.4 h to 45 d with larger inter-frequency spacing for higher ranges, as described in the Methods. This is similar to frequency spacing increases (usually logarithmic) in most wavelet-based research studies across many fields¹⁻¹⁰. The time resolution of the data is 1 h, based on intrinsic device limitations for detection count storage, and a lower frequency limit was therefore set near the Nyquist frequency of 2 h. The higher frequency limit of 45 d was set based on the availability of three months of data in some subjects (per inclusion criteria) and the fact that accurate power and phase estimates require two full-cycles due to edge effects (minimum amount of data is 90 days, thus maximum period length is 45 days).

Supplementary Fig. shows analyses of subselected data (3-6 months) for several of the data-sets. Often components at longer periodicities are enhanced. To sustain the hypothesis of a

specific long-term rhythmicity it seem important to show that several subselected periods from the same patient give similar long-terms peaks of periodicity.

As the Reviewer states, selected raw data separated by several years are already shown for the subjects with more than two years of data in Supplementary Fig. 4. In some cases, the relative stability of the multidien rhythmicity is visually apparent in the raw data (see, for example, subject S10). However, we quantify this by including an autocorrelation analysis in Supplementary Fig. 5e, which confirms relative stability over long durations of recording. We believe this analysis is a more rigorous approach than comparing periodograms from arbitrarily selected epochs.

2. The orange curve in Fig. 1f remains a poor estimate of the daily counts trace since higher frequencies are missing. Supplementary Fig. 2 helps, but does not seem to correspond to the period shown in Fig. 1f, which might be improved by showing another critical higher frequency component.

The orange curve indeed represents only one component of the daily counts trace, and it alone cannot capture all features of the raw signal. Wavelet analysis does not fit a single sinusoidal wave to complex data, but rather it decomposes the data into a number of component oscillations. Our intent with the original rendering of Fig. 1 was to highlight a central finding of this work by showing only the strongest slow (multidien) oscillation present in the raw data. However, we have now revised Fig. 1 to show the two main multidien oscillations (10-day and 26-day periodicities) for this subject. We also bring a concept from Supplementary Fig. 2 to the forefront of the paper by showing in Fig. 1f how combining a broader range of wavelet coefficients allows for faithful data reconstruction with excellent goodness-of-fit ($r = 0.93$).

3. The reply seems to miss the question. Was a single (orange) waveform used as the basis to extract a phase relation of seizure to interictal frequency? How does the phase relation change if a better, possibly multiple-wavelet fit was used?

We apologize for the lack of clarity. We believe we have now addressed this concern with edits to the Methods and revisions to Fig. 1. Phase analysis was performed on the circadian period and the shortest multidien period ($\pm 33\%$ to account for some variation in period length, as described in Methods). It is likely that combining different multidien rhythms could further improve seizure risk prediction, as the Reviewer posits. In fact, we show that combining circadian and multidien frequencies (two-dimensional phase analysis) does improve seizure risk prediction (Supplementary Fig. 8). However, determining the combination of periodicities that optimally explains seizure timing is beyond the scope of this initial study as it would require multi-dimensional (3-D, 4-D, or more) phase incorporation and potentially novel statistical approaches. This question remains the subject of active investigation.

6. This reply also misses the question. Fig. 4b suggest that ictal events occur when interictal frequency is increasing, whether the period is 10, 20 or 30 days. If so, there is a simple way to check the data and also to support the idea that some seizure-related state parameter depend

on interictal frequency. Seizures should occur only when the day-to-day interictal frequency is increasing. This approach would not ignore the circadian fluctuations, since day-to-day frequency may increase over several consecutive days. Presumably ictal events would be very probable.

This question is closely related to Point 4 raised by Reviewer 2; please refer to that paragraph for a complete answer. We agree that dynamic seizure prediction methods are an important application of our findings. However, we anticipate that, based on the nature of the data, successful approaches will be more complex than the Reviewer suggests. Day-to-day IEA changes may not necessarily show monotonic increases or decreases that parallel the underlying multidien oscillation(s) due to influences from noise and potentially weaker oscillation frequencies, just as day-to-day changes in a stock market index may not reflect long-term market oscillations. Therefore, day-to-day changes in IEA alone would be too noisy to derive consistent relationships to seizure timing. Wavelet decomposition is a well-established method for analysis of time series data¹⁻¹⁰, and the various circular statistics used throughout our analysis serve to check the data by revealing the significance of the results.

Reviewer # 4:

The authors have done extensive modifications and been very responsive to the criticisms. This paper now is much more convincing, and is a result that I am very interested in. It has novel analysis and unprecedented data. I have just a few comments that need clarification.

IEA need to be defined earlier and explicitly. There should be something in Results right after “Subjects” that defines it briefly. And it should again be described explicitly (i.e. “for this device, IEA is defined by....”).

We now include the following explicit definition in the first paragraph of Results: “For this study, IEA is defined as hourly rates of detections of epileptiform discharges using subject-specific algorithms designed by clinicians (Supplementary Fig. 1a).”

The problem is that IEA in Neuropace is essentially a proprietary definition. And additionally it is unique for every patient, and can be changed by the user. This is a bigger deal than the authors make it—it is likely that some of the other reviewers of this paper actually misunderstood this as well. The reader is most likely going to believe it means either a traditional spike wave discharge and/or a clear electrographic seizure. And the reader is going to believe that you are claiming to have detected all of them. Thus it is imperative to give an objective definition, followed by an explanation of the data that are saved.

Supplementary Fig. 1a shows examples of the various patterns of activity (i.e. not just spike-waves) that can be detected by the RNS System. Together with the more explicit definition of IEA which we now include (see response to previous point), we hope to avoid misleading any readers.

It is true that RNS System detection algorithms are customizable and patient-specific. However, we describe in the Methods that blocks of data with stable detection settings were analyzed separately and normalized (z-score) before concatenation with other blocks. Fluctuations in IEA can therefore be compared across epochs independent of the sensitivity and specificity of detection settings within those epochs. We do not claim to detect all interictal epileptiform discharges, and in fact we acknowledge in the Discussion that changes in detection parameter sensitivity can affect absolute detection counts.

The other problem is that the authors themselves are not consistent when referring to IEA. The definition of IEA versus “IEA rates used in the analysis” are ambiguous and used interchangeably. The abstract states that seizures are “preferentially occurring near the peak of IEA.” I think you mean peak of IEA rates? (the “peak of the IEA” would be the instant the spike is maximum amplitude). There is also a section in results called “Phase analysis of IEA peak...” It is not the IEA peak, but the peak of the IEA rates. Another section reads “underlying IEA rhythms” Again, you are not analyzing the IEA rhythm, but the periodicity of the peak rates. This style needs to be policed throughout the whole paper.

We have now added early in the manuscript (Abstract and Introduction) a definition of Interictal Epileptiform Activity (IEA) as the rate of interictal epileptiform discharges (i.e. hourly counts of detections stored by the RNS System), not the waveforms of individual epileptiform discharges. We have revised the manuscript for consistency and clarity in this terminology.

In Supp Fig 1a, a statement should be added reminding the reader that these data are not used in the analysis, just the counts of these data.

We have added the following line to the Legend for Supplementary Fig. 1a: “Hourly counts of detections of these patterns, not the waveforms themselves, constitute the raw data analyzed in this study.”

I suggest a simple change to line 63: “wavelet transform to resolve component rhythmicity of the IEA rate”. This is because many readers will assume you are decomposing the actual IEA waveforms, since that is what is almost invariably done with wavelets and EEG.

We have revised this line in accordance with our clarified definition of IEA (see above).

I also suggest a stronger “sell” of Suppl Fig 2—this is really a very nice figure to help understand what is going on.

We now preface the manuscript’s reference to Supplementary Fig. 2 with “signal processing steps depicted in detail in...”

Supp Fig 8 and Fig 5 are beautiful. I’ve never seen data like this before—this should be very impactful, as it shows some patients that clearly have variability of seizure risk on two different

time scales. Remarkable! But I do not think the authors explained these results well enough in the text. It is somewhat buried in a jargon-rich section titled "Phase-space analysis of seizure risk". I strongly suggest making this more of a highlight of the paper, accentuating it with more user-friendly title and explanation, and probably adding it to the abstract. A section title along the lines of "seizure risk variability on circadian and multidien time scales" (or anything like that) would be helpful, as would a written presentation that is more practical and less buried in details. I suspect this result will be the one that is most often cited in this paper by the general audience (who are not going to be familiar with "phase-space"). Making it easier to find and understand is important.

We appreciate these positive comments highlighting the novelty of our data. We have revised the Abstract to emphasize these results, and we have changed the section title to "Seizure risk modulation on circadian and multidien time scales".

The new analysis on risk ratios is very helpful to see which patients have discernable changes.

We identified typographical errors in Fig. 5 and Supplementary Fig. 8 and have changed "OR" to "RR".

Typo line 101: "across in these subjects"

We have corrected this typographical error.

REFERENCES

1. Fontolan, L., Morillon, B., Liegeois-Chauvel, C. & Giraud, A.L. The contribution of frequency-specific activity to hierarchical information processing in the human auditory cortex. *Nat Commun* **5**, 4694 (2014).
2. Zhang, Z., Telesford, Q.K., Giusti, C., Lim, K.O. & Bassett, D.S. Choosing Wavelet Methods, Filters, and Lengths for Functional Brain Network Construction. *PLoS One* **11**, e0157243 (2016).
3. Gadhomi, K., Lina, J.M. & Gotman, J. Discriminating preictal and interictal states in patients with temporal lobe epilepsy using wavelet analysis of intracerebral EEG. *Clin Neurophysiol* **123**, 1906-1916 (2012).
4. Richard, C.D., *et al.* SWDreader: a wavelet-based algorithm using spectral phase to characterize spike-wave morphological variation in genetic models of absence epilepsy. *J Neurosci Methods* **242**, 127-140 (2015).
5. Faust, O., Acharya, U.R., Adeli, H. & Adeli, A. Wavelet-based EEG processing for computer-aided seizure detection and epilepsy diagnosis. *Seizure* **26**, 56-64 (2015).
6. Ortiz-Rosario, A., Adeli, H. & Buford, J.A. Wavelet methodology to improve single unit isolation in primary motor cortex cells. *J Neurosci Methods* **246**, 106-118 (2015).
7. Moberg, A., *et al.* Highly variable Northern Hemisphere temperatures reconstructed from low- and high-resolution proxy data. *Nature* **433**, 613-617 (2005).

8. Cazelles, B., Chavez, M., Magny, G.C., Guegan, J.F. & Hales, S. Time-dependent spectral analysis of epidemiological time-series with wavelets. *J R Soc Interface* **4**, 625-636 (2007).
9. Ramsey, J.B. Wavelets in Economics and Finance: Past and Future. *Studies in Nonlinear Dynamics & Econometrics* **6**(2002).
10. Cazelles, B., *et al.* Wavelet analysis of ecological time series. *Oecologia* **156**, 287-304 (2008).

REVIEWERS' COMMENTS:

Reviewer #2 (Remarks to the Author):

The authors have addressed my remaining concerns.
I recommend publication.

Reviewer #3 (Remarks to the Author):

This resubmitted ms from Baud and colleagues suggests there is a multi-day and an intra-day rhythmicity in the timing of interictal events in epileptic patients with epilepsy. Ictal events seem to occur on a specific phase of the multi-day rhythms. These are useful observations based on very long-term monitoring. I found the responses to previous points were not very persuasive, even if I'm not sure that should preclude publication.

1. I asked about the form of the multi-day oscillation. Specifically the example shown in Fig. 1 seems to show a relatively sudden (2-4 days) increase in the frequency of hourly counts followed by a much slower decay (10-15 days). Similar pattern may be present in data of Supp Fig. 3 and in S1M, S3M, S7M, S10M... of Supplementary Fig. 4. Possibly it informs on biological processes involved. The wavelet approach, while statistically correct, may not adequately extract information on the shape of the multi-day oscillation. Its disappointing that the authors do not go further.

2. Similarly the authors do not really answer the question on the relation between timing of seizure events and the multi-day oscillation, even if the stock market analogy is very amusing.

Reviewer #4 (Remarks to the Author):

The authors have answered my questions. There was one line that was still ambiguous, though. Line 69 "hourly counts of IEA detections" should probably just say " IEA rate" or something like that.

NCOMMS-17-03166B -- RESPONSES TO REVIEWERS' COMMENTS

Reviewer #2:

*The authors have addressed my remaining concerns.
I recommend publication.*

Reviewer #3:

This resubmitted ms from Baud and colleagues suggests there is a multi-day and an intra-day rhythmicity in the timing of interictal events in epileptic patients with epilepsy. Ictal events seem to occur on a specific phase of the multi-day rhythms. These are useful observations based on very long-term monitoring. I found the responses to previous points were not very persuasive, even if I'm not sure that should preclude publication.

1. I asked about the form of the multi-day oscillation. Specifically the example shown in Fig. 1 seems to show a relatively sudden (2-4 days) increase in the frequency of hourly counts followed by a much slower decay (10-15 days). Similar pattern may be present in data of Supp Fig. 3 and in S1M, S3M, S7M, S10M... of Supplementary Fig. 4. Possibly it informs on biological processes involved. The wavelet approach, while statistically correct, may not adequately extract information on the shape of the multi-day oscillation. Its disappointing that the authors do not go further.

We now acknowledge this point in the Discussion and suggest it as a topic for future investigation: "Further analysis of the rise and decay kinetics of IEA fluctuations may be informative with regard to underlying biological mechanisms."

2. Similarly the authors do not really answer the question on the relation between timing of seizure events and the multi-day oscillation, even if the stock market analogy is very amusing.

The Reviewer previously stated, "Seizures should occur only when the day-to-day interictal frequency is increasing." In fact, our results indicate that it is easily possible for a seizure to occur on a day when IEA is lower than the previous day, provided that the day of the seizure falls on the rising phase of a multi-day oscillation. Thus, we maintain that phase information regarding multi-day IEA oscillations explains seizure timing better than day-to-day fluctuations in IEA.

Reviewer #4:

The authors have answered my questions. There was one line that was still ambiguous, though. Line 69 "hourly counts of IEA detections" should probably just say "IEA rate" or something like that.

We have made the suggested change in the manuscript text.